# Wind tunnel investigation of the aerodynamic response of two 15 MW floating wind turbines

Alessandro Fontanella[1], Alan Facchinetti[1], Simone Di Carlo[1], and Marco Belloli[1]

[1]Mechanical Engineering Department, Politecnico di Milano, Milano, Via La Masa 1, 20156, Italy.

**Correspondence:** Alessandro Fontanella (alessandro.fontanella@polimi.it)

**Abstract.** The aerodynamics of floating turbines is complicated by large motions which are permitted by the floating foundation, and the interaction between turbine, wind and wake is not yet fully understood. The object of this paper is a wind tunnel campaign finalized at characterizing the aerodynamic response of a 1:100 scale model of the IEA 15 MW subjected to imposed platform motion. The turbine aerodynamic response is studied focusing on thrust force, torque and wake at 2.3D downwind the rotor. Harmonic motion is imposed in the surge, sway, roll, pitch and yaw directions with several frequencies and amplitudes, which are selected to be representative of the two 15 MW floating turbines developed within the COREWIND project. Thrust and torque show large-amplitude oscillations with surge and pitch motion, which main effect is an apparent wind speed; oscillations in thrust and torque are negligible with the other motions, which main effect is to alter the wind direction. The thrust and torque response measured in the experiment is compared with predictions of a quasi-steady model, often used for control-related tasks. The agreement is good in case of low-frequency surge motion, but some differences are seen in the pitch case. The quasi-steady model is not predictive for the response to wave-frequency motion, where blade unsteadiness may take place. Wake was measured imposing motion in five directions with frequency equal to the wave frequency. The axial speed is slightly lower with motion compared to the fixed case. The turbulence kinetic energy is slightly lower too. Wave-frequency motion seems to produce a more stable and lower flow mixing.

## 1 Introduction

Floating offshore wind turbines (FOWTs) have numerous advantages over their bottom-fixed counterpart when it comes to harnessing the wind power resource of deep-sea sites, which make up a significant portion of the offshore wind resource. Wind speed is generally higher in these regions which are farther from the coast, but bottom-fixed foundations are not cost effective when the water depth is higher than 50 m, and installation could be easier as most operations can be done in a port. Technical feasibility has been proved by a first wave of pre-commercial projects deployed worldwide, and the second phase of offshore floating wind power is underway with the first pilot floating wind farms being developed in these years (Barter et al. (2020)). Installed floating wind power is expected to grow significantly over the next few years, but there are still many technical challenges to be solved to make this possible. Floating wind has been included by Veers et al. (2019) among the open research questions in the science of wind energy. The large dimensions of modern machines combined with the additional degrees of freedom of floating foundations give rise to new interactions between the turbine, the wind and wakes, which are not yet fully

understood. Most studies about the design and response of floating wind turbines implicitly assume the aerodynamic analysis methods developed for bottom-fixed turbines are valid also in the floating case (Sebastian and Lackner (2013)), so data is needed to evaluate the capabilities of such models.

In this sense, scale model experiments which focus is the aerodynamic response in floating turbines, like the one covered in this article, are useful in two ways: to gain knowledge about the physics of the process, and for producing datasets for codes validation. Farrugia et al. (2014) carried out experiments with the scale model of a TLP turbine subjected to different wave and turbine operating conditions, and analyzed the effect of wave-induced motions on the turbine power output and wake comparing results for the floating case to the bottom-fixed condition. With the support of a free vortex wake code, it is shown that platform motion causes fluctuations in the aerodynamic torque, a reduced mean power coefficient, and a time-varying tip-vortex transport velocity. Rockel et al. (2014) measured the wake of a model wind turbine with particle image velocimetry (PIV) with platform pitch motion and compared it with four wake models. The wake structure has been found to be more complex with rigid-body motions and this requires developing improved wake models. Fu et al. (2019) conducted wind tunnel experiments to understand the effect of platform pitch and roll motion on the power output and wake of a model turbine. It is shown the wake is significantly altered by imposed motion, and the power fluctuations exhibit a marked peak in the spectral content in correspondence of the frequency of motion. Bayati et al. (2016) conducted a wind tunnel campaign with a scale model of the DTU 10MW subjected to imposed surge and pitch motion and compared thrust measurements to a BEM model with dynamic wake. This experiment was complemented by a second (Bayati et al. (2017b)) focused on the effects of imposed surge motion on the wake. It is seen the wake axial velocity has fluctuations at the frequency of motion and the amplitude of these oscillations depends on the average thrust coefficient, and the frequency of motion. The findings of these two experiments, and the lack of clear conclusions, promoted the UNsteady Aerodynamics of FLOating Wind turbines (UNAFLOW) project which goal was to study the aerodynamic response of a FOWT subjected to large surge motion, covering blade forces, rotor-integral forces, and wake. The methodology and experimental results of the experiment are discussed in the articles of Bayati et al. (2018a), Bayati et al. (2018b), and Fontanella et al. (2021). Among the project results, it is found the turbine thrust response is quasi-static for reduced frequency up to 0.5, the near-wake energy content is increased in correspondence of the frequency of motion, and the travel speed of the tip-vortex has periodic oscillations. Schliffke et al. (2020) studied the wake of a FOWT with imposed surged motion by means of a porous-disk model placed in an atmospheric boundary layer wind tunnel. Results show the turbulence intensity in the far-wake is lower for a floating turbine compared to the bottom-fixed case, and the spectral content of the axial velocity has a peak at the frequency of imposed motion. The same porous-disk model was used by Garcia et al. (2022) to analyze the behavior of the wake center of a floating wind turbine subjected to imposed surge motion. Stereoscopic PIV measurements show that surge motion has very small effects on the far-wake center position, but the frequency content of the wake has a clear trace of the platform motion.

As said, experimental data are useful for validation of aerodynamic codes. The dataset of the UNAFLOW experiment is currently examined in the Phase III of the OC6 project, where experimental data are compared to numerical tools which are based on different principles and have a variable fidelity level. Previous efforts are the work of Cormier et al. (2018), which used the UNAFLOW data to assess predictions of a BEM model, a free vortex wake model, and a blade-resolved computational

fluid dynamics (CFD) model, and of Mancini et al. (2020) which focused on rotor forces and extended the comparison to an actuator line model. Another example is the article of Ribeiro et al. (2022) where the UNAFLOW data of rotor forces and wake are used to validate a free-wake panel method.

The research in this work investigates the aerodynamic response of two 15 MW floating turbines developed in the COREWIND project (Mahfouz et al. (2021)). An experimental testing campaign has been conducted at the Politecnico di Milano wind tunnel with a 1:100 model of the IEA 15 MW turbine which was subjected to imposed platform motion so to simulate the rigid-body movement of a floating turbine. The main contributions of this work are as follows.

– Previous experiments have shown that platform motion affects the turbine aerodynamic response, primarily the thrust force, the rotor torque (and power) and the turbine wake, but conclusions are still partial. In this sense, we decided to investigate the above mentioned quantities with motion in the surge, sway, roll, pitch and yaw directions. The imposed motion is sinusoidal and in one direction at a time, so it is still idealized, but the test matrix is defined to be representative of the motion of a 15 MW floating turbine in a realistic deployment site. To this end we considered the Activefloat and WindCrete floating turbines developed in the COREWIND project with reference to the Gran Canaria site (Mahfouz et al. (2021)). The sinusoidal motion was preferred over a more realistic one, with the platform moving simultaneously in all directions and in a broad frequency range, to ease future comparisons with numerical codes. The motion of a FOWT is large in the low-frequency range, where resonant excitation occurs, and in the wave-frequency range, where hydrodynamic loading associated with waves is large. The effect of motion on the turbine aerodynamic response should be more pronounced at these frequencies, which are covered by the motion conditions of the experiment.

– The aerodynamic thrust and torque is often introduced in control-oriented models of floating turbines by means of the static power and thrust coefficients and, when the model is linearized, by means of their gradient with respect to blade pitch, rotor speed and wind speed. Here, we try to assess if and when the modeling approach based on the static power and thrust coefficients is effective, comparing its predictions with the thrust and torque response to surge and pitch motion.

– The wind tunnel measurements of Bayati et al. (2018a), Bayati et al. (2018b), and Fontanella et al. (2021) focused on the wake-flow response with low-frequency surge motion. In this campaign wake was measured with motion in five directions with a typical wave frequency. Wake is measured in non-turbulent inflow conditions to highlight the effect of turbulence produced by the floater motion on the flow mixing and wake recovery.

The foreseen impact of this paper is as follows.

– The database of force measurements can be used for validation of numerical codes. Recent validation tasks focused on the force response to low-frequency surge and pitch motion. With data of this campaign, the comparison can be extended to higher frequencies and to other directions of motion. Moreover, one goal of the COREWIND project is to use a combination of hybrid hardware-in-the-loop experiments in the wind tunnel (Belloli et al. (2020)) and in the wave basin (Battistella et al. (2018)) for assessing the response of two 15 MW floating turbines in operating and extreme conditions.

In wave basin hybrid experiments aerodynamic loads are simulated with a force actuator and an aerodynamic model of the turbine. Wind tunnel measurements of the force response can be used for tuning this model. Having calibrated the aerodynamic part of the wave basin experiment on the response of the turbine model should favor the comparison of hybrid wave basin results with results of hybrid wind tunnel experiments that use the same turbine model.

– Comparison of wind tunnel results with the control-oriented model of the thrust and torque response to turbine motion gives an idea of where this is valid and where it may have some shortfalls.

– With the recent progress of the floating turbine technology, studies are emerging about the response and control of floating wind farms, like the one about loads and wake meandering of Wise and Bachynski (2020) and the one about vertical wake steering of Nanos et al. (2021). Wind tunnel measurements of the wake inflow increase knowledge about wakes of floating turbines, and this is of utmost importance for pushing further the floating wind technology.

The structure of the remainder of this paper is as follows. Section 2 describes the testing facility, the turbine scale model and the measurements that were carried out. The load cases of the experiment, and the rationale behind their definition, are discussed in Sect. 3. Section 4 presents the results about rotor forces with fixed and moving turbine, and here experimental results with surge and pitch motion are compared to quasi-static linear model. Results about the turbine wake with different types of platform motion are reported in Sect. 5. Graphs of the results section are made in accordance with the recommendations of Stoelzle and Stein (2021) which should make plots easier to read; information in line graphs is coded with line type, perceptually uniform color maps are used for 2D plots, and the red-grey-blue color map is used to underline data direction in 2D plots with zero midpoint. Finally, Sect. 6 discusses some conclusions and tries to give some suggestions for future work about the aerodynamic response of floating turbines.

## 2 Experimental campaign

The testing activity was carried out in the atmospheric boundary layer test section of the Politecnico di Milano wind tunnel (GVPM), which has dimensions: 13.84 m wide x 3.84 m high x 35 m long. The test setup is shown in Fig. 1, and a sketch of the wind tunnel facility is reported in Appendix B. The turbine was mounted on a 6-DOFs robotic platform to enable forced motion.

### 2.1 Scale turbine design and specifications

The wind turbine is a 1:100 scale model of the IEA 15 MW (Allen et al. (2020)). The turbine rotor was scaled to preserve the power ($C_P$) and thrust ($C_T$) coefficients of the reference turbine despite the reduced size and a wind speed reduction factor of 3. The blade design is carried out to match the lift distribution along the span while preserving the tip-speed ratio, similarly to what was done by Bayati et al. (2017a) for a 1:75 model of the DTU 10 MW. A single airfoil, the SD7032 (the polars measured in 2D experiments are reported in the article of Fontanella et al. (2021)), is used along the blade, the chord is increased with respect to the reference full-scale rotor preserving the original distribution, and the twist distribution is altered

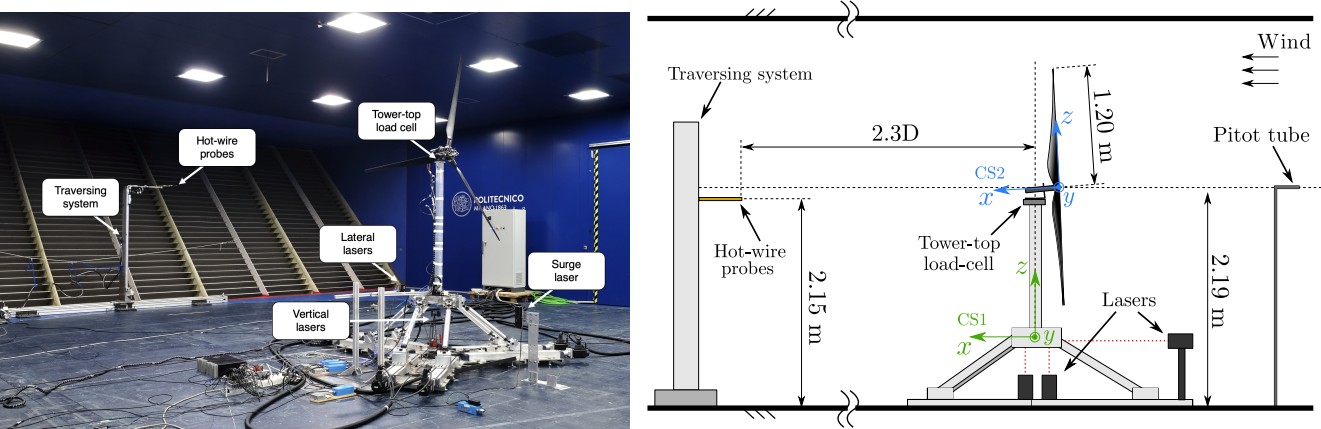

**Figure 1.** Experimental setup in the Polimi wind tunnel (left) and schematic of the test setup with the coordinate systems (CS) used for measurements and their analysis (right).

to have the target non-dimensional lift force. The turbine tower, of 75 mm diameter, is rigid since we focus on the effect of rotor motion associated with platform motion rather than with tower deformability. The turbine has collective blade-pitch control and variable-speed generator control. The blades were built to be rigid to exclude any aeroelastic interaction, which was outside of the scope of this research. The main properties of the turbine model are summarized in Table 1.

**Table 1.** Key parameters of the wind turbine model.

| Parameter | Unit | Value |
|---|---|---|
| Rotor diameter | m | 2.400 |
| Blade length | m | 1.110 |
| Hub diameter | m | 0.180 |
| Rotor overhang | m | 0.139 |
| Tilt angle | ° | 5.000 |
| Tower-to-shaft | m | 0.064 |
| Tower diameter | m | 0.075 |
| Tower length | m | 1.400 |
| Nacelle mass | kg | 1.975 |
| Blade mass | kg | 0.240 |
| Rotor mass | kg | 2.041 |
| Tower mass | kg | 2.190 |

## 2.2 Measurements

The measurements taken in the tests are shown in Fig. 1 and are: interface forces between tower-top and nacelle by a 6-components load cell; platform position by laser transducers; 3-components ($U$, $V$, and $W$) wind velocity in the wake by hot-wire probes. The two probes were moved in the cross-wind direction (Y) of CS1, from $-1.6$ m to $+1.6$ m with a discretization of 100 mm, at a fixed distance X = 2.3D, and Z = 2.15 m. The undisturbed wind velocity was measured by a pitot tube 7.15 m upstream of the turbine rotor, centerline, and hub height (not visible in the picture of Fig. 1). Additional technical details about the hot-wire system and the load-cell are given in Appendix B. All measurements were sampled synchronously at 2 kHz and stored at model scale.

## 3 Definition of load cases

The experiment considered two functioning conditions for the wind turbine, reported in Table 2. In one, the wind turbine is operated at the blade pitch and tip-speed ratio (TSR) that give the maximum power coefficient. A fine-trim search shows the maximum power coefficient is attained for TSR = 9.0 and $\beta = -3.5°$, and these values were selected for WC1. The rotor was designed to have its maximum efficiency at TSR = 9.0 and $\beta = 0°$ as the IEA 15 MW, and the fact this occurs for a lower $\beta$ may be due to errors in the blade mounting. In WC2 the rotor speed is equal to its rated value, and the blade pitch is trimmed to reduce the rotor efficiency. The TSR is equal to the steady-state value for the IEA 15 MW at the corresponding wind speed, and $\beta$ is adjusted to match the power coefficient of the reference turbine. Also in WC2 the pitch offset with respect to the IEA 15 MW is found to be -3.5°. Active turbine control was not used, and in all tests the blade pitch angle and rotor speed were constant. Tests were performed using an empty inlet configuration (i.e., without roughness elements or turbulence generators) for a uniform inflow velocity and a resulting turbulence intensity around 2%.

**Table 2.** Wind turbine operating conditions considered in the experiment ($\beta$ is collective pitch).

| Condition | Wind speed [m/s] | Rotor speed [RPM] | TSR [-] | $\beta$ [°] |
|---|---|---|---|---|
| Wind condition 1 (WC1) | 2.95 | 210 | 9.0 | -3.5 |
| Wind condition 2 (WC2) | 5.00 | 216 | 5.4 | 8.5 |

For the cases with motion, the turbine was forced to oscillate alternatively in the surge, sway, roll, pitch, and yaw degree-of-freedom (DOF). As it is noticed by Sebastian and Lackner (2013), the platform motions modify the operating environment for the turbine rotor compared to the bottom fixed case, mostly by altering the apparent wind speed perceived by the rotor disk. Different movements have a different effects:

- surge, pitch, and yaw move the rotor disk in the wind direction altering the magnitude of wind speed. In the surge case, the additional wind speed due to motion is constant across the rotor, in the pitch case it increases linearly with height, with yaw the increment is linear with radial distance and of opposite sign on the left and right side of the rotor;

- pitch, roll, and yaw introduce an effective wind shear;

- sway, heave, and roll move the rotor in the cross-wind direction and modify the angle formed by wind with the rotor axis, creating a skewed (i.e., non-axial) inflow.

When the wind is constant and uniform in space, the effect of sway and heave in terms of apparent wind perceived by rotor is similar: one inclines the velocity vector in the horizontal plane, and the other in the vertical plane. At the same time, the wind tunnel section is large compared to rotor ($A_{\mathrm{rotor}}/A_{\mathrm{tunnel}} = 0.08$) but its height is comparable to the rotor diameter ($D/h_{\mathrm{tunnel}} = 0.62$) and this is cause of anisotropic blockage. In reason of these two considerations, the turbine was moved only in the sway direction.

In general, the motion of a FOWT is due to wind and wave excitation. In mild waves, the motion response is driven by wind and second-order hydrodynamic loads, at it is mainly at the natural frequencies of the rigid-body modes; the amplitude of motion at wave frequency depends on the strength of waves and on the wave direction. For these reasons, the motion for every DOF was imposed at three frequencies: the natural frequency of the mode for the Activefloat and for the WindCrete, and the frequency of the wave spectrum peak for Gran Canaria site. The motion frequencies are summarized in Table 3.

**Table 3.** Full-scale and model scale values of the natural frequencies and wave frequency for the Activefloat and WindCrete. Full-scale values are taken from (Mahfouz et al. (2021)).

| FOWT | Surge | Sway | Roll | Pitch | Yaw | Wave |
|---|---|---|---|---|---|---|
| WindCrete full-scale [Hz] | 0.012 | 0.012 | 0.024 | 0.024 | 0.092 | 0.111 |
| Activefloat full-scale [Hz] | 0.006 | 0.006 | 0.031 | 0.031 | 0.012 | 0.111 |
| WindCrete model scale [Hz] | 0.350 | 0.350 | 0.700 | 0.700 | 2.625 | 3.175 |
| Activefloat model scale [Hz] | 0.175 | 0.175 | 0.875 | 0.875 | 0.375 | 3.175 |

Reduced frequency is a dimensionless number used to characterize the degree of unsteadiness of an aerodynamic system which is caused by a harmonic perturbation in the flow. The rotor reduced-frequency $f_r$ is often used for describing the rotor-level unsteadiness associated with the global response of the rotor disk and its wake. It is defined as:

$$f_r = \frac{f_m \mathrm{D}}{U_\infty},$$
(1)

where $f_m$ is the frequency of motion, $U_\infty$ the free-stream wind speed, and D the diameter of the turbine rotor. The reduced frequency of the load cases of the experiment is shown in Fig. 2. Results of the testing of Fontanella et al. (2021) indicate surge motion causes minimal unsteady aerodynamic behavior when $f_r$ is lower than 0.5. In the experiment discussed in this paper $f_r$ was increased up to 3 to verify if the conclusions of Fontanella et al. (2021) are valid for motion at the wave frequency.

At the same time, blade-level unsteadiness may occur, as predicted for example by Theodorsen theory, when the blade-reduced frequency $f_c$ is high. The blade reduced-frequency $f_c$ is defined as:

$$f_c = \frac{\pi f_m c}{V}, \tag{2}$$

where $c$ is the chord of a blade section and $V$ is the velocity it perceives. When $f_c$ is small the circulatory contributions to airfoil lift from Theodorsen's theory dominate; when $f_c$ is high, the apparent mass contributions, which arise from flow acceleration effects, begins to dominate, and flow unsteadiness is expected to take place. As explained by Sebastian and Lackner (2013), unsteadiness typically occurs for $f_c \geq 0.05$, that is when the apparent mass effects due to local flow acceleration become meaningful. By means of the limit on $f_c$ it is possible to define a threshold frequency for turbine motion $f_{m,th}$ beyond which airfoil-level unsteadiness may occur. This is obtained from Eq. 2 with $f_c = 0.05$, and $V$ equal to the peripheral speed of the blade section:

$$f_{m,th} = \frac{0.05\sqrt{U_\infty^2 + (r\omega_r)^2}}{\pi c}, \tag{3}$$

where $\omega_r$ is the rotor speed and $r$ is the radial position of a blade section. The threshold frequency for the turbine scale model and the two operating conditions of the experiment is reported in Fig. 2. The shaded area in the figure corresponds to the operating range of the IEA 15 MW. Motion with frequency that falls to the right of the curve may cause blade-level unsteadiness. The blade aerodynamic response is quasi-steady for motion at the natural frequencies of the two floating turbines. Motion with frequency of the WindCrete yaw mode and at the wave frequency may result in some unsteadiness for the blade sections with $r/R < 0.5$, and so in blade-level and rotor-level unsteady aerodynamics occurring together. Only rotor-level unsteadiness is expected at the other frequencies of motion.

Equations 1-2 account for the motion frequency, but not its amplitude. When the motion frequency is high, the rotor or blade aerodynamics may be different than the quasi-steady prediction, but the effects of unsteadiness may be small if the motion amplitude is small.

The aerodynamic rotor loads are expected to be linearly proportional to the rotor apparent wind, as it is found by Fontanella et al. (2021) and Mancini et al. (2020) for the case of low-frequency surge motion. In this test campaign the three frequencies $f_m$ were tested with two values of amplitude for any platform DOF; in the case of motion in surge, pitch and yaw directions, which creates an apparent wind, this is done to confirm the linearity of aerodynamic loads with respect to the apparent wind (i.e. to motion amplitude when the frequency is fixed). Two amplitudes were tested also in the case of sway and roll motion to cover a wider range of conditions. The motion amplitudes $A_m$ were defined with the following rules:

– surge: to produce a normalized maximum velocity $\Delta U/U_\infty = 0.04\text{-}0.05$, where $\Delta U = 2\pi f_m A_m$. The resulting amplitude values are between 0.010 m and 0.180 m (1.0-18.0 m full-scale). The values of $\Delta U/U_\infty$ are chosen to be similar to those used in the campaign of Fontanella et al. (2021) to facilitate comparison with the results of that test;

– pitch: to have a normalized maximum velocity at hub height $\Delta U_{\mathrm{hh}}/U_\infty = 0.04\text{-}0.05$, where $\Delta U_{\mathrm{hh}} = 2\pi f_m A_m d_{\mathrm{hub}}$, and $d_{\mathrm{hub}}$ the distance between the rotor apex and the center of pitch rotation. With this choice, the apparent wind speed at

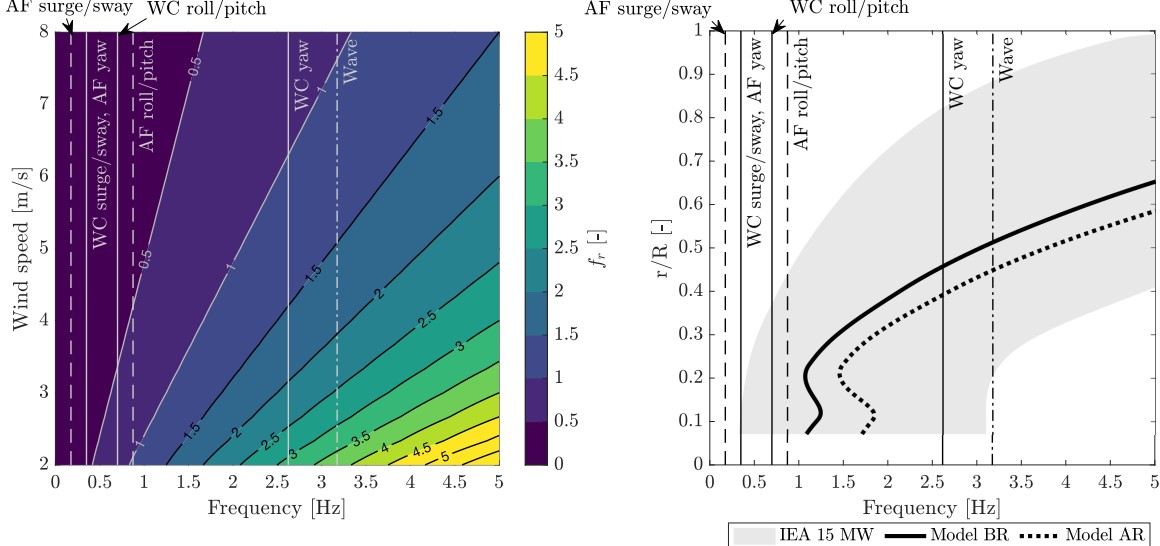

**Figure 2.** Left: rotor reduced-frequency as a function of motion frequency and wind speed; the reduced frequencies of the load cases are identified by markers (WC1 is wind condition 1, WC2 wind condition 2). Notice the heave frequency of the two platforms is not shown because heave motion was not investigated. Right: the threshold motion frequency beyond which blade-level unsteady aerodynamics may occur for the turbine model is compared to the motion frequencies of the experiments (AF is Activefloat, WC is WindCrete); the shaded area covers the threshold frequency for the IEA 15 MW in its operating range.

the hub due to motion is the same of surge cases. The main effect of surge and pitch motion is to alter the wind speed perceived by the rotor, so having the same $\Delta U_{hh}$ should favor comparisons between the surge and pitch load cases. The pitch motion is between 0.25-2.76 deg;

– yaw: to give a normalized maximum velocity of the rotor edge $\Delta U_R/U_\infty = 0.03, 0.05$, with $\Delta U_R = 2\pi f_m A_m R$. In this way, the wind speed perceived by the outermost sections of the blade is similar to surge cases. The resulting yaw motion has amplitude of 0.3-3.0 deg;

– sway: to give a maximum wind misalignment with respect to rotor axis $\alpha = 2°, 4°$, where $\alpha = \tan^{-1}(2\pi f_m A_m/U_\infty)$. The amplitude of motion is in the range 0.011-0.064 m (1.1-6.4 m full-scale);

– roll: to have a maximum wind misalignment at hub height $\alpha_{hh} = 2°, 4°$, where $\alpha = \tan^{-1}(2\pi f_m A_m z_{hh}/U_\infty)$, and $z_{hh}$ is the vertical distance between rotor apex and the roll axis. The wind misalignment is the same of sway cases, and the amplitude of motion is of 0.4-3.0 deg.

The complete test matrix obtained with the rules above is reported in Appendix A. The values of amplitude and frequency of the surge and the pitch motion are comparable to those considered by Ramos-García et al. (2021) to investigate the aerodynamic response of the IEA 15 MW mounted on the WindCrete floater by means of two different aerodynamic solvers. Wake

measurements are carried out for motion conditions with wave frequency. These were chosen in place of the low-frequency conditions because the wake response to wave-frequency motion was not covered by any previous experiments, which instead focused on the low-frequency motion.

## 4 Results about rotor forces

The global response of a FOWT is influenced by rotor-integral loads which are often identified in the combined thrust force and torque of the three blades (e.g., by Lemmer et al. (2020)). The torque is strictly connected to the turbine power output, the dynamics of the drivetrain and the controller response. In a wind turbine, power is extracted from wind at the expense of a thrust force exerted on the rotor, which results in the rigid and flexible motion of the structure in the along-wind direction (van der Veen et al. (2012)). Thrust and torque are state-dependent because the motion of the structure produces an apparent wind which affects the rotor loads. Hereafter, we report and discuss the experimental results in terms of these rotor forces.

The thrust force and torque, expressed in the CS2 reference frame which is non-rotating and fixed to the rotor hub, are obtained projecting the force measurements of the tower-top load cell. Results for the fixed turbine are reported in terms of steady-state power and thrust coefficients. Results with imposed motion is the spectral content at the frequency of motion of the dynamic thrust and dynamic torque that are obtained with different types of motion. The thrust/torque response with surge and pitch is compared to a linear quasi-steady model, which is often used for control purposes.

### 4.1 Fixed turbine

Tests were run with fixed turbine and the two wind conditions of Table 4 to characterize the turbine aerodynamic response without motion. The resulting power and thrust coefficients are reported in Table 4 together with those of the IEA 15 MW in the corresponding operating conditions. The scaled rotor performs close to the full-scale turbine; the thrust coefficient, which primarily depends on the distribution of normal force along the blade, is closely matched. As discussed by Wang et al. (2021), the axial velocity in the wake is largely set by the rotor thrust, and the wake of the turbine scale model is representative of the full-scale turbine if the thrust coefficient is the same. Some mismatch is instead seen for the power coefficient. This is largely influence by airfoil efficiency, which is lower for the turbine scale model.

**Table 4.** Steady power ($C_P$) and thrust ($C_T$) coefficient for the wind turbine model (subscript "WTM") and for the IEA 15 MW (subscript "ref") in the operating conditions of the experiment (see Table 2).

| Condition | $C_{P,\mathrm{WTM}}$ | $C_{P,\mathrm{ref}}$ | $C_{T,\mathrm{WTM}}$ | $C_{T,\mathrm{ref}}$ |
|---|---|---|---|---|
| | [-] | [-] | [-] | [-] |
| WC1 | 0.35 | 0.49 | 0.78 | 0.77 |
| WC2 | 0.13 | 0.17 | 0.20 | 0.20 |

In addition to tests with the wind conditions of Table 4, the blade pitch-TSR maps of the power and thrust coefficient were measured in steady wind of $U_\infty$ = 4 m/s, with a resolution for TSR of 0.3 and 3° for blade pitch. The maps are reported in Fig. 3. The values of power and thrust coefficient obtained from interpolation of the maps for the WC2 are similar to those measured in the experiment with WC2 wind. The same is not true for WC1, and values obtained from the maps are higher than those measured in the experiment at 3 m/s. The aerodynamic behavior of the blade is sensitive to the Reynolds number, which

is lower in WC1 than with the 4 m/s for which the maps were obtained. The lower Reynolds results in a slightly lower lift force and higher drag, and so in a lower power coefficient and slightly lower thrust coefficient.

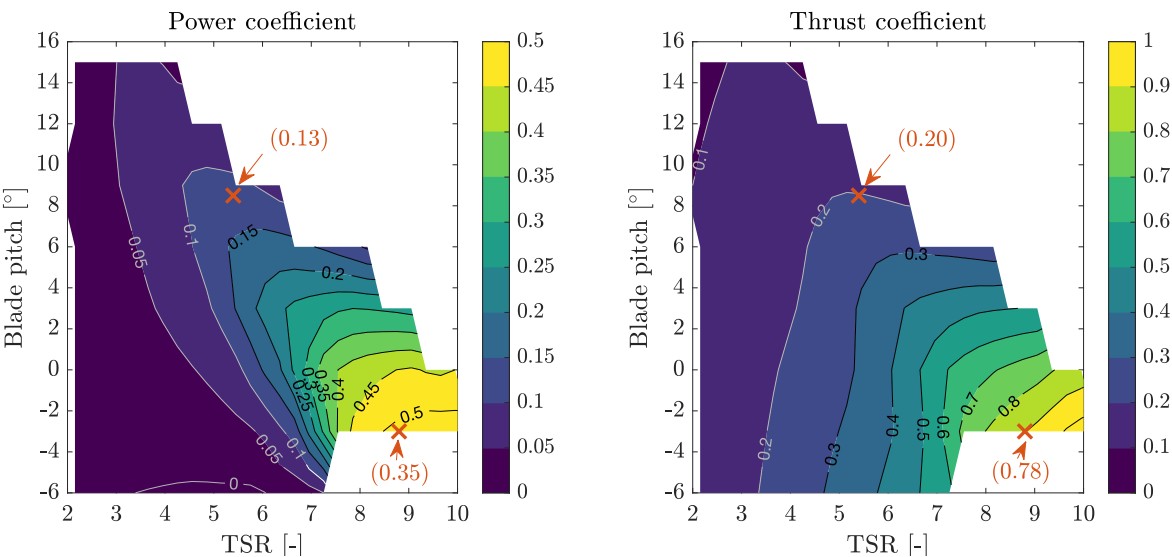

**Figure 3.** Map of the power and thrust coefficients of the turbine model measured in steady wind with $U_\infty$ = 4 m/s. The × marks correspond to the turbine operating conditions of Table 4, and the coefficient values obtained in tests with fixed turbine are reported in brackets.

## 4.2    Moving turbine

Force measurements with moving turbine are processed as depicted in Fig. 4 in order to remove the contribution of the rotor-nacelle inertia associated with the motion of the structure. For every load case two tests are run imposing the same motion to

the turbine base: one with no wind and fixed rotor, and one with wind and spinning rotor. Measurements of the two tests are windowed so to have the same integer number of periods of the imposed motion. Time series of the 6 tower-top forces from the no-wind tests are subtracted from the time series of the test with wind. In doing this operation, we assume the flexible response of the turbine is small (i.e., the model is rigid) and equal with and without wind; we also assume the aerodynamic force developed by the blades is small in the test without wind. The forces obtained with the force subtraction procedure

are projected to CS2. The focus of the analysis is the thrust/torque response at the frequency of platform motion. Hence, we

compute the fast Fourier transform and we look at the amplitude and phase of the harmonic component with frequency equal to the frequency of motion.

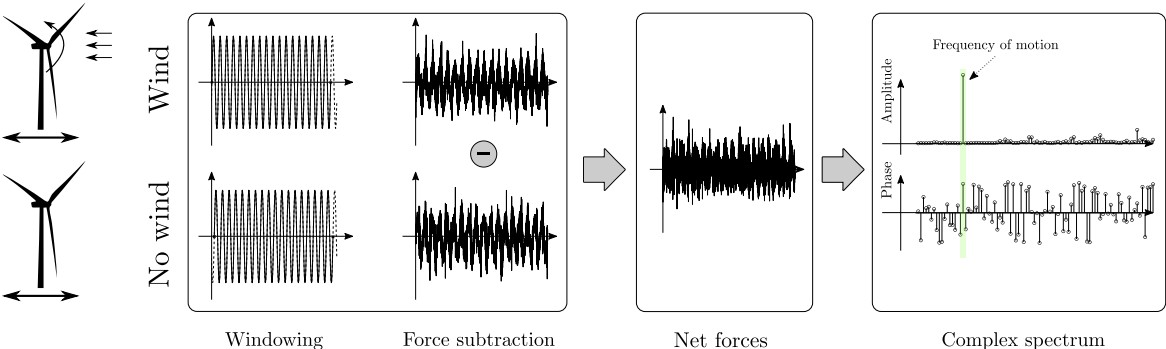

**Figure 4.** Scheme of the post-processing applied to force measurements in tests with moving turbine. Two tests are carried out for every motion condition, one with wind and spinning rotor, one without wind and with still rotor. Time series from the two tests are windowed so to have the same integer number of motion periods. Then, forces from the test with no wind are subtracted from forces with wind. The resulting forces are examined taking the complex spectrum and studying the harmonic content at the frequency of the imposed motion (amplitude and phase).

The harmonic components of thrust and torque at the frequency of imposed motion and for all the wind and motion conditions we studied are shown in Fig. 5 in terms of amplitude and phase-shift with respect to motion. The same y-axis limits are used for all motion directions to highlight the motion conditions that have the largest influence on the rotor force response. For thrust, the oscillations with the largest amplitude are observed with surge and pitch motion. For motion of equal frequency and the same normalized variation of wind speed, the amplitude of oscillations is larger in WC2 than in WC1: the amplitude of thrust oscillations depends on the operating condition. With the other motions, oscillations are small, up to 0.19 N, and of the same order of magnitude of the response to the inflow turbulence. Motions in directions other than the wind direction do not affect the turbine thrust in a significant way. Torque oscillations are of meaningful amplitude with surge, pitch, sway and roll. For surge cases with low-frequency motion, the phase shift of thrust and torque is equal to $-\pi/2$, but with wave-frequency motion, the phase-shift is slightly higher. The amplitude with pitch motion is similar to surge motion for equal amplitude of hub displacement. However it is more difficult to see a trend in the phase shift, which is not $-\pi/2$. The thrust and torque response with motion in the surge and pitch directions is examined with more detail in the Sect. 4.2.1. The amplitude of torque oscillations is large also seen with roll and sway. The amplitude is about linearly proportional to the frequency of motion and the phase shift is around $-\pi$; the torque maximum is when the acceleration due to platform motion is directed as the blade peripheral speed. The large amplitude of torque variations, which is not matched by thrust, and their phase shift suggest the torque response is mostly due to mechanical inertia which could not be removed by the force post-processing. This is due to the inability to completely lock the rotor in the tests without wind, which resulted in small oscillations of the rotor of less than 10° of amplitude. In principles, the mechanical inertia of the rotor could be estimated also with tests with spinning rotor, but

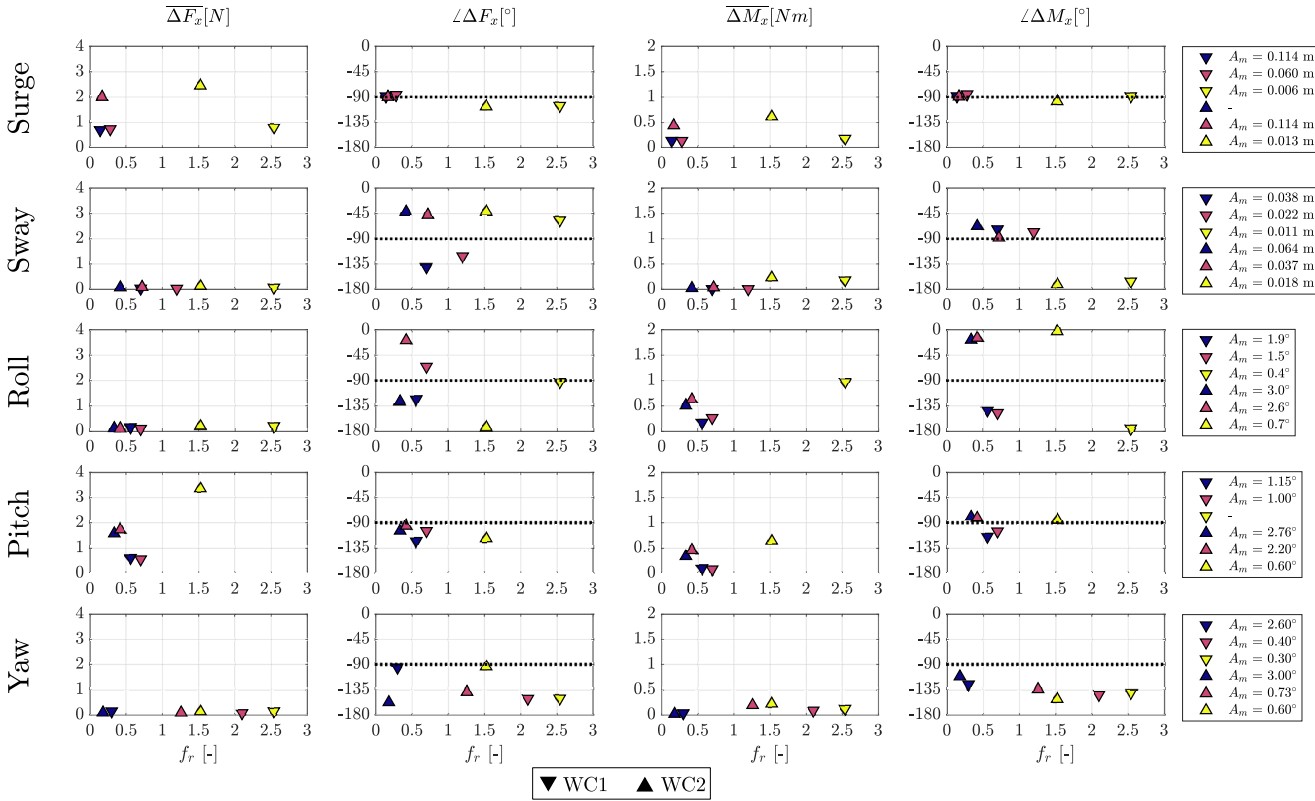

**Figure 5.** Amplitude and phase shift of the dynamic thrust force ($\overline{\Delta F_x}$, $\angle \Delta F_x$), and amplitude and phase shift of the dynamic torque ($\overline{\Delta M_x}$, $\angle \Delta M_x$) as function of reduced frequency $f_r$. Reversed-triangle markers correspond to results for the wind condition 1 (WC1), triangle markers to results for wind condition 2 (WC2), and colors identify different values of the motion amplitude $A_m$.

these must be run in the void, and it is unfeasible given the size of the turbine model. Given the test procedure adopted in this campaign it is not possible to isolate the oscillations of aerodynamic torque due to unsteady wind only, however we can reasonably say these are small enough to be masked by the uncertainties associated with the testing.

### 4.2.1 Quasi-static model of thrust and torque

The thrust and torque response to surge and pitch is compared with the prediction of a linear quasi-static model, which is often used when dealing with control of floating turbines. The surge and pitch DOFs are usually included in control-oriented models (e.g., those of van der Veen et al. (2012), Pegalajar-Jurado et al. (2018), Fontanella et al. (2020), Lemmer et al. (2020)) because these show the largest response amplitude among all the platform motions when wind and waves are aligned. Generally, in such models the aerodynamic rotor thrust is introduced as

$$F_x = \frac{1}{2}\rho C_T(\omega_r, \beta, U)A_r U^2\,, \tag{4}$$

where $\rho$ is the air density, $C_T$ the thrust coefficient, $A_r$ the rotor area, and $U$ the wind speed; and the aerodynamic torque as

$$M_x = \frac{1}{2}\rho C_Q(\omega_r, \beta, U)A_r R U^2 \,, \tag{5}$$

where $C_Q$ is the torque coefficient, and $R$ is the rotor radius. The first-order linearization of Eq. 4 and Eq. 5 is

$$F_x \simeq F_{x,0} + \left.\frac{\partial F_x}{\partial \omega_r}\right|_0 (\omega_r - \omega_{r,0}) + \left.\frac{\partial F_x}{\partial \beta}\right|_0 (\beta - \beta_0) + \left.\frac{\partial F_x}{\partial U}\right|_0 (U - U_0) \,, \tag{6}$$

$$M_x \simeq M_{x,0} + \left.\frac{\partial M_x}{\partial \omega_r}\right|_0 (\omega_r - \omega_{r,0}) + \left.\frac{\partial M_x}{\partial \beta}\right|_0 (\beta - \beta_0) + \left.\frac{\partial M_x}{\partial U}\right|_0 (U - U_0) \,, \tag{7}$$

where $(\cdot)_0$ denotes the steady-state value of a quantity for a given turbine operating point. For the experiments with imposed motion, the blade pitch and rotor speed are constant, $U$ is the apparent wind speed for the rotor, and $U_0 = U_\infty$. Equations 6-7 become

$$300 \quad F_x \simeq F_{x,0} - \dot{x}_{\text{hub}}\left.\frac{\partial F_x}{\partial U}\right|_0 \,, \tag{8}$$

$$M_x \simeq M_{x,0} - \dot{x}_{\text{hub}}\left.\frac{\partial M_x}{\partial U}\right|_0 \,, \tag{9}$$

where $\dot{x}_{\text{hub}}$ is the hub velocity. With harmonic motion, the variation of thrust force and torque with respect to the steady-state value is

$$305 \quad \Delta F_x(t) = -K_{U,T}(2\pi f_m)A_{\text{hub}}\sin(2\pi f_m t - \pi/2) \,, \tag{10}$$

$$\Delta M_x(t) = -K_{U,Q}(2\pi f_m)A_{\text{hub}}\sin(2\pi f_m t - \pi/2) \,, \tag{11}$$

with $K_{U,T} = \partial F_x/\partial U|_0$, $K_{U,Q} = \partial M_x/\partial U|_0$, and $A_{\text{hub}} = A_m$ in case of surge motion, and $A_{\text{hub}} = A_m h_{\text{hub}}$ for pitch motion. The phase of the force response with respect to motion is $-\pi/2$. The zero-peak amplitude normalized by amplitude of hub

motion is

$$\overline{\Delta F_x}/A_{\text{hub}} = K_{U,T}(2\pi f_m) \,, \tag{12}$$

$$\overline{\Delta M_x}/A_{\text{hub}} = K_{U,Q}(2\pi f_m) \,, \tag{13}$$

so it is proportional to the frequency of motion by the slope of the thrust/torque with respect to wind speed, evaluated at the

315 steady-state operating point. This approach is referred to as quasi-steady theory (QST) because it predicts the aerodynamic

response in dynamic wind conditions caused by platform motion based on the aerodynamic response at steady state (i.e., the $C_T$ and $C_P$ maps of Fig. 3). This linearized aerodynamic model is widely used in floating turbine control. One example is the paper of van der Veen et al. (2012), where he uses this approach to explain the negative-damping phenomenon associated with pitch control in above rated wind for floating turbines. Abbas et al. (2022) used it to introduce an additional feedback term in the pitch controller in order to decouple the platform pitch and the rotor dynamics and stabilize the system.

Figure 6 compares the thrust and torque response to surge and pitch motion from the experiment and from the QST model of Eq. 12-13. The sensitivities $K_{U,T}$ and $K_{U,Q}$ are computed from the thrust and torque coefficients respectively. The expressions of $K_{U,T}$ as function of $C_T$ and of $K_{U,Q}$ as function of $C_Q$ are derived computing the derivative of Eq. 4-5 with respect to the wind speed $U$

$$K_{U,T} = \left. \frac{\partial F_x}{\partial U} \right|_0 = \left( \rho C_T A_r U \right)_0 + \left( \frac{1}{2} \rho A_r U^2 \frac{\partial C_T}{\partial U} \right)_0, \tag{14}$$

$$K_{U,Q} = \left. \frac{\partial M_x}{\partial U} \right|_0 = \left( \rho C_Q A_r \mathrm{R} U \right)_0 + \left( \frac{1}{2} \rho A_r \mathrm{R} U^2 \frac{\partial C_Q}{\partial U} \right)_0. \tag{15}$$

In a more compact form, Eq. 14-15 are

$$K_{U,T} = \left. \frac{\partial F_x}{\partial U} \right|_0 = \frac{F_{x,0}}{U_\infty} \left( 2 - \left. \frac{\partial C_T}{\partial \lambda} \right|_0 \frac{\lambda_0}{C_{T,0}} \right), \tag{16}$$

$$K_{U,Q} = \left. \frac{\partial M_x}{\partial U} \right|_0 = \frac{M_{x,0}}{U_\infty} \left( 2 - \left. \frac{\partial C_Q}{\partial \lambda} \right|_0 \frac{\lambda_0}{C_{Q,0}} \right), \tag{17}$$

where $\lambda$ is the TSR. $K_{U,T}$ and $K_{U,Q}$ depend solely on the turbine operating condition and the consequent steady-state response, so Eq. 16-17 are evaluated for both the operating conditions of the experiment: $F_{x,0}$ and $M_{x,0}$ are those of fixed-turbine tests (see Table 4), the partial derivatives of the rotor coefficients are obtained from the gradient of the rotor coefficients map shown in Fig. 3, with $C_Q = C_P/\lambda$. The $C_T$ and $C_Q$ coefficients were measured for TSR increments of 0.3, which is similar to the variation of TSR caused by surge and pitch motion in the load cases of the experiment (0.16-0.29). Figure 3 shows the power coefficient and, to a lesser extent, the torque coefficient depend on the wind speed, and values measured at 3 m/s are not coincident with those predicted by the $C_P$ and $C_T$ maps that were obtained at 4 m/s. In Eq. 16-17 we assume the derivatives of the $C_T$ and $C_Q$ do not depend on the wind speed. This assumption is a source of uncertainty for the QST model, for torque more than for thrust, because the aerodynamic torque is more sensitive to Reynolds number.

In Fig. 6 we see the QST model predicts the aerodynamic response to low-frequency surge motion, but not at wave frequency. At low frequency the amplitude of aerodynamic loads is linear with frequency and the phase-shift with respect to motion is $-\pi/2$. The agreement between QST and the experiment is better for WC1 compared to WC2. The QST model is sensitive to the accuracy of the static coefficients from which it is derived; the difference between the slope of the linear fit to experimental data and the QST can be due to the interpolation of $(\partial C_T/\partial \lambda)_0$ and $(\partial C_Q/\partial \lambda)_0$ which is required to evaluate the derivatives for the a pitch angle of $8.5°$. The good agreement with QST in case of surge motion is in line with the findings of Fontanella

et al. (2021) where the same result was obtained for low-frequency motion of a 1:75 scale model of the DTU 10 MW. The point at the wave frequency shows a response amplitude higher than the linear trend, and the phase shift is lower than $-\pi/2$. The QST is not predictive for this condition: a significant portion of the blade is likely interested by blade-level unsteady aerodynamics for a motion frequency equal to wave frequency ("Wave" in Fig. 2) and this may explain the non-quasi-static response of this condition. Also in the case of pitch motion, the amplitude of torque and thrust is linear with frequency except for the point at wave frequency, which shows a higher response amplitude than what is predicted by QST. In case of pitch, unlike what happens with the surge, the phase of the aerodynamic loads measured in the experiment is never $-\pi/2$. As it is shown in Fig. 2, blade-level unsteadiness can be the cause of the different phase shift only in the case of pitch motion with wave frequency but is not a valid explanation for cases with low-frequency motion. The rotor reduced-frequency is higher for low-frequency pitch motion compared to surge ($2\times$ for the WindCrete and $5\times$ for the Activefloat) and this may give some rotor-level unsteadiness. Another reason for the different phase behavior may be the skewed inflow created by pitch motion. Surge and pitch are equivalent in terms of the apparent wind speed at hub height, but pitch motion produces a velocity gradient across the rotor height, with the upper portion feeling a higher wind speed than the lower one. This is not accounted for by the QST model, which models the rotor as a single point coincident with the hub.

To sum up, experimental data show the QST model is valid for the thrust/torque response to low-frequency surge motion. Instead, we observe some differences for higher-frequency motion, as it occurs as a consequence of wave excitation, and for pitch motion in general.

## 5 Wake measurement results

This section describes results about hot-wire measurements. Focus of the analysis is the effect of platform motion at wave frequency on the wind velocity in the wake.

Figure 7 shows the average wake deficit normalized by the free-stream velocity $U_\infty$ at X $= 2.3$D for different types of motion. In a wind farm perspective, the wake deficit defines the inflow conditions for downstream turbines. The wake shape is set by the turbine operating condition, and shows a double-gaussian profile in WC1 cases, and a gaussian profile in WC2 cases. In both the wind conditions the shape is not symmetric with respect to the rotor axis, and velocity is generally lower for negative-Y. This behavior is also seen in the wake measurements of Fontanella et al. (2021), which are with a different rotor and in the same wind tunnel. As it is observed by van der Hoek et al. (2022), the asymmetry in the wake can be due to the vortices shed by the tower interacting with the rotor wake. Another possible reason is the anisotropic blockage, which may hinder the wake expansion on one side more than the other. The velocity at the wake extremities is 17% higher than the free-stream velocity in WC1 and WC2. This difference is partially explained as the effect wind tunnel blockage (according to the model of Glauert (1935) the overspeed due to blockage is 3% for WC1 and 1% for WC2) and most of it may be due to an offset between the hot-wire probes and the upstream pitot tube which is used to take the measurement of $U_\infty$.

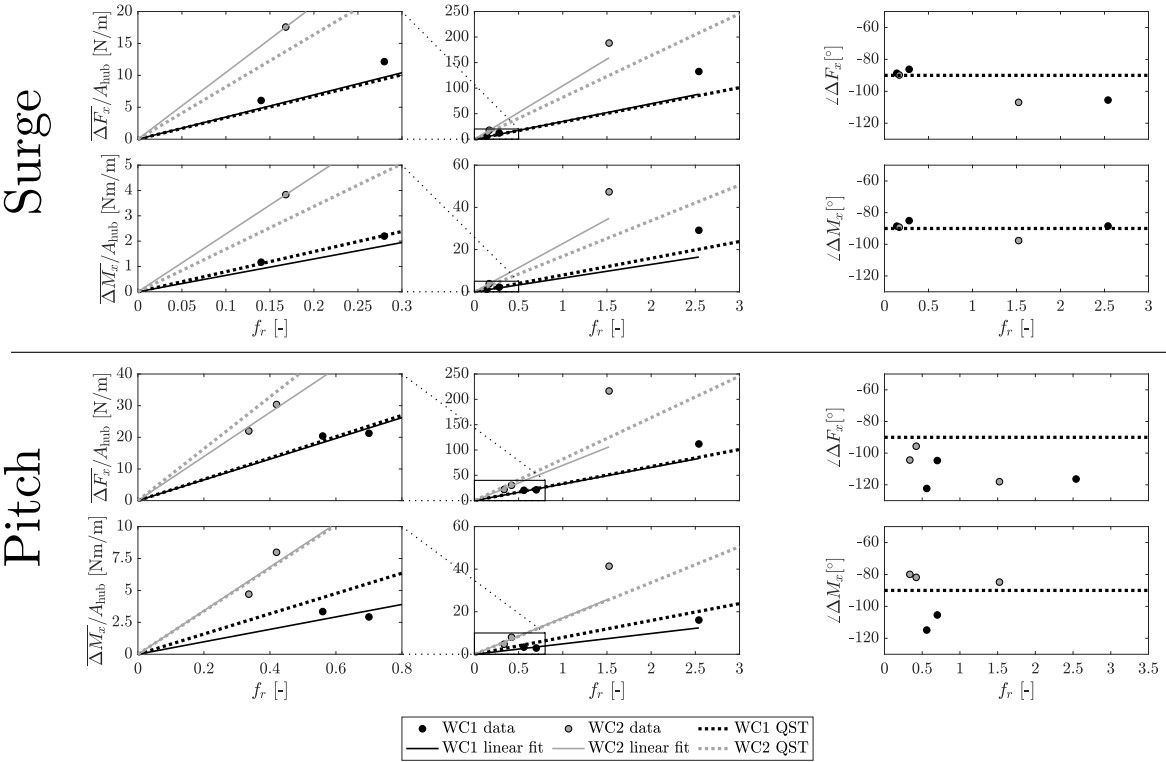

**Figure 6.** The thrust and torque responses to surge and pitch motion in the two wind conditions of the experiment WC1 and WC2 is compared to quasi-steady theory (QST) in terms of normalized zero-peak amplitude and phase shift with respect to platform motion. The linear fit is the least-square linear regression of amplitude data which is obtained excluding the points at the highest reduced frequency and intercepts the axes in the origin; QST is the prediction of the quasi-steady theory model computed from the steady-state thrust and torque coefficient maps of Fig. 3.

The wake with motion is compared to the fixed turbine case computing the average wake deficit for the rotor area, defined as:

$$D_{\mathrm{avg}} = \frac{1}{U_\infty} \left( \frac{\sum_{i=1}^{N} |y_i| U_i}{\sum_{i=1}^{N} |y_i|} \right), \tag{18}$$

with $y_i = -1.2, \ldots, 1.2$, and $N = 25$. The results are reported in Table 5 and show that the average velocity across the rotor is slightly lower with wave-frequency motion compared to the fixed case. In a wind farm perspective, this means the energy in the flow available for a hypothetical floating turbine at 2.3D from the upstream unit working in fully-waked condition would be slightly less than in the bottom-fixed case. The bottom panels of Fig. 7 show the wake deficit increment with motion compared to the bottom-fixed case for different Y positions. This information might be used to compute the change in the radial distribution of aerodynamic loads for a waked floating turbine. In WC1, the velocity is lower in correspondence of the

outer sections of the rotor, and increased outside it; the variation is about the same regardless of the type of motion. In WC2, the velocity is decreased between $Y = \pm(0.5\text{-}1)$ m, and the largest decrement is with sway, pitch, and yaw motion. Sway has the lowest difference for WC1 but the highest difference for the WC2.

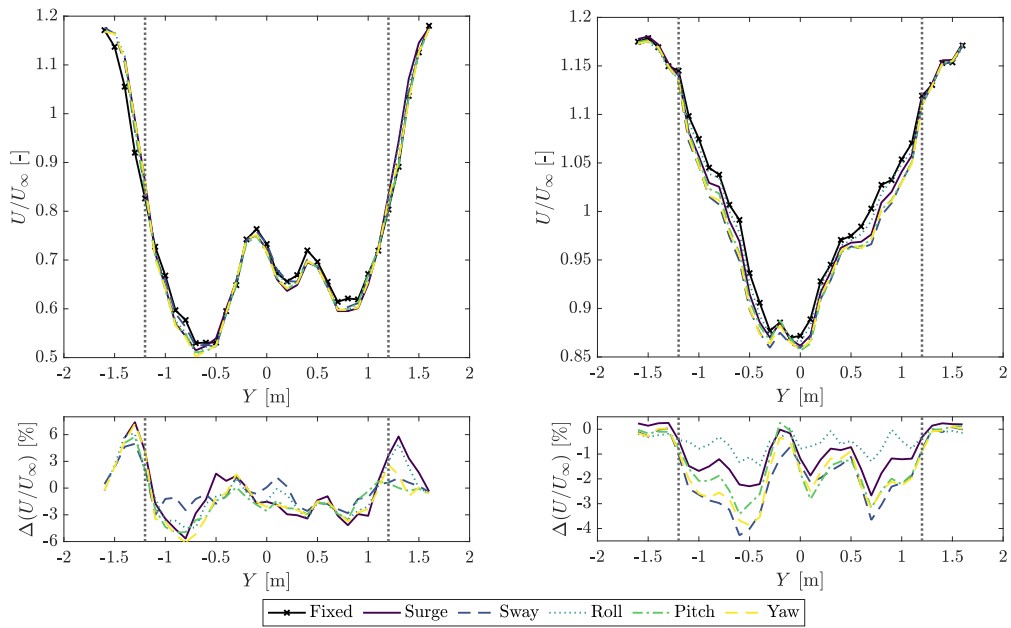

**Figure 7.** Average wake deficit at $X = 2.3D$ for the fixed case and with different motions at the wave frequency, in WC1 (top left) and WC2 (top right). $\Delta(U/U_\infty)$ is the wake deficit increment with motion with respect to the fixed case (WC1: bottom left; WC2: bottom right). The vertical dotted lines mark the edge of the rotor.

**Table 5.** Average wake deficit for the rotor area for the two wind conditions WC1 and WC2 and different motions at the wave frequency. $\Delta$ is the percent change with motion with respect to the fixed case.

| Condition | WC1 | $\Delta$WC1 | WC2 | $\Delta$WC2 |
|---|---|---|---|---|
| | [-] | [%] | [-] | [%] |
| Fixed | 0.667 | - | 1.034 | - |
| Surge | 0.660 | $-1.079$ | 1.021 | $-1.300$ |
| Sway | 0.661 | $-0.884$ | 1.011 | $-2.310$ |
| Roll | 0.659 | $-1.319$ | 1.028 | $-0.609$ |
| Pitch | 0.657 | $-1.604$ | 1.015 | $-1.876$ |
| Yaw | 0.657 | $-1.589$ | 1.013 | $-2.117$ |

Ramos-García et al. (2021) and Fu et al. (2019) observed the wake recovery for a FOWT which undergoes to pitch motion with a frequency close to the platform pitch mode is different than in the bottom-fixed case. In this case flow mixing is higher for a floating turbine because increased turbulence due to low-frequency motion promotes a faster break down of the strong vortex structures. In this sense, Fig. 8 complements Fig. 7 by showing the turbulence kinetic energy (i.e., $k = \frac{1}{2}(\sigma_u^2 + \sigma_v^2 + \sigma_w^2)$, where $\sigma_i$ is the variance of the i-th velocity component: $u$ axial, $v$ transverse, $w$ vertical) at $X = 2.3D$ for the fixed turbine

and with different types of platform motion at the wave peak frequency. The distribution of $k$ about the Y axis is given by the turbine operating condition and is consistent for the fixed and floating scenario. In the WC1, most of the turbulence kinetic energy is concentrated around the edge of the rotor, and is associated with tip vortices. In the WC2, $k$ is maximum between $Y = -0.5$-0 m, which corresponds to the wake center, and the peak is likely associated with root vortices. $k$ is non-symmetric, and is more pronounced at negative-Y. With motion, $k$ is generally lower than for the bottom-fixed case, except at the rotor

edge for the WC2 case. The high-frequency platform oscillations caused by response to wave forcing seems to produce a stronger and more stable wake. The lower flow mixing makes the wake recover more slowly than without motion. This agrees with what was observed by Ramos-García et al. (2021) for pitch motion at 0.057 Hz (i.e.,1.622 Hz at model scale). The authors noticed that for lower motion frequencies differences between a bottom-fixed and floating turbine are significant for $x/D > 5$, but for higher frequencies the wake recovery is independent of the downstream location.

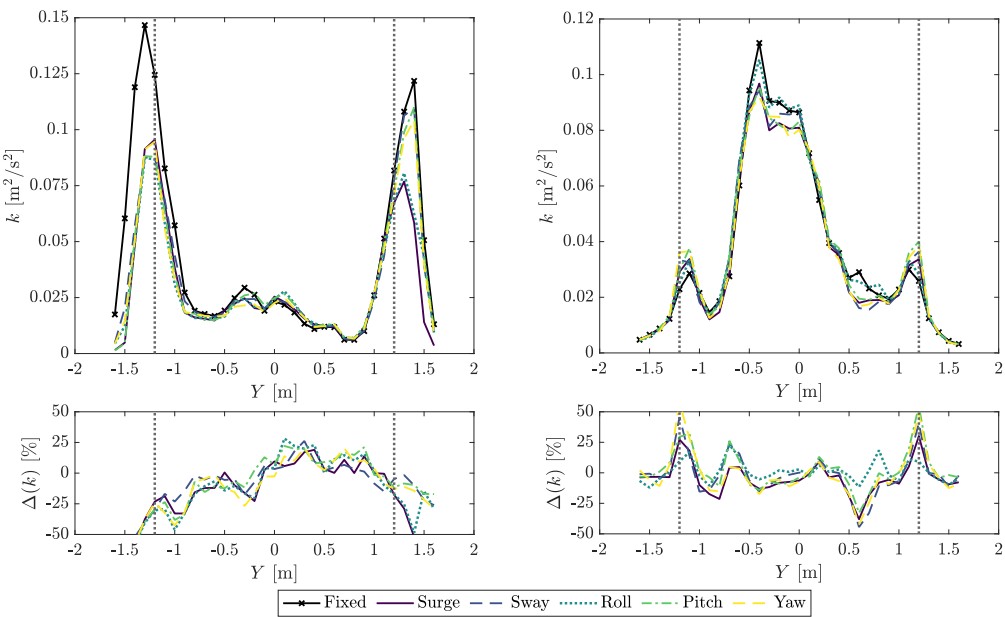

**Figure 8.** Turbulence kinetic energy at $X = 2.3D$ for the fixed case and with different motions at the wave frequency, in WC1(left) and WC2 (right). $\Delta(k)$ is the turbulence kinetic energy increment with motion with respect to the fixed case (WC1: bottom left; WC2: bottom right). The vertical dotted lines mark the edge of the rotor.

In a floating turbine, motion-induced fluctuations of the wind velocity in the wake are expected to happen for different reasons. With surge and pitch, these occur due to the dynamic inflow created by the motion and the oscillating thrust force. With motion in the other directions, the magnitude of the wind speed perceived by the rotor is similar to the fixed case, and oscillations of the axial velocity in the wake are explained as the effect of wake meandering: the velocity at a downstream location varies periodically because the wake is moved laterally and vertically. This occurs also with pitch and is superposed to the effect of dynamic inflow. The presence and relevance of velocity oscillations in the wake due to motion is assessed from the phase-averaged wind velocity in the turbine wake. The time series of the three wind speed components in 94 motion cycles at the point Y are $u, v, w(t = 0, \ldots, 94T_m, Y)$, where $T_m = 1/f_m$. The time series are windowed to isolate the 94 motion periods and phase-averaged obtaining $\overline{u}, \overline{v}, \overline{w}(t = 0, \ldots, T_m, Y)$. This operation preserves the signal content which has the same periodicity of motion and filters out the rest. The mean velocity in each point ($U(Y)$ shown in Fig. 7, $V(Y)$, $W(Y)$) is subtracted from the phase-averaged data. As a final step, the dependence from the motion period $T_m$ is replaced with the motion phase $\phi = (t/T_m) \cdot 360°$. This analysis is carried out for every Y position and produces the space-time evolution of phase-averaged variation of wake velocity components due to platform motion. Figure 9 shows the results for WC2, at $X = 2.3D$, motion at the wave frequency, and in five directions. Results for WC1 are omitted because the amplitude of velocity oscillations is larger in WC2 compared to WC1. In WC1 the amplitude of axial-velocity fluctuations associated with motion is small, less 1% of $U_\infty$, and similar in magnitude to the turbulence in the wake of the fixed turbine. Moreover, it is easier to interpret results for WC2 compared to WC1 because in WC2 the wake symmetry with respect to the X axis is better. The effect of motion in the surge direction is visible in the axial and vertical velocity components. In the first half of the motion cycle, the axial velocity is increased across the entire wake, and it is decreased in the second half; the variation is stronger on the left of the average wake center position compared to the right. The variation in the wake velocity is the effect of the perturbation introduced by the rotor when it moves, that is propagated downstream. Thus, the phase with respect to motion of the velocity variation depends on the propagation speed and the distance from rotor. For the vertical component, the peak velocity is reached at different times by sections in different Y positions; the peak is reached first on the left side and then it propagates towards the right. Motion in the sway and roll directions clearly affects all three velocity components. The axial velocity is increased periodically across the entire wake, but, differently than with surge, the peak in the outer sections is delayed moving from the wake center to the wake edges. We applied the tracking method described by Coudou et al. (2018) to the axial velocity data to detect the position of the wake center, and this does not change significantly in one sway/roll cycle. If the wake core moves, the motion is smaller than the spatial resolution of wake data (i.e., 100 mm, or 10 m full-scale). Sway and roll excite in a significant manner the transverse and vertical velocity; the response is stronger in the central portion of the wake and negligible on the edges; the maxima and minima travel periodically from left to right. Since no significant meandering is detected, the velocity oscillations in the central portion of the wake can be the result of the interaction of root vortices that are generated when the turbine operates in WC2. The effect of pitch motion on the axial velocity is similar to surge, but slightly more pronounced. Differently than with surge, pitch motion has a strong effect on the vertical velocity, which is periodically increased and decreased across the wake width. This supports the idea that oscillating pitch moves the wake up and down in the X-Z plane. The response to oscillating yaw is visible mostly in the axial velocity and in a narrow region in the center of the wake. Here the velocity is increased and decreased

periodically. There are no visible effects on the other two components. The wake-center tracking algorithm does not detect any motion of the wake center also in the case of yaw motion. From these results, it seems that high-frequency yaw oscillations like those caused by waves do not move the wake center in a significant way, contrary to the case of static (or quasi-static) yaw, which is often used for redirecting the wake laterally (Meyers et al. (2022)). As for sway and roll motion, the velocity oscillations in the center of the wake suggest a possible interaction between the root vortices.

## 6  Conclusions

Wind tunnel testing has been conducted to characterize the aerodynamic response of a 1:100 scale model of the IEA 15 MW subjected to harmonic platform motion. The turbine is forced to move in five motion directions; for every type of motion we considered different combinations of amplitude and frequency, selected to be representative of the dynamic response of the two 15 MW floating turbines of the COREWIND project.

The rotor response to platform motion is examined with focus on rotor-integral loads, thrust and torque. Surge and pitch motion move the turbine rotor in the wind direction altering the apparent wind speed. As a consequence, thrust and torque show large-amplitude oscillations. With yaw motion, the apparent wind is periodically increased on one side of the rotor and decreased on the other, and the apparent wind is higher at the rotor periphery compared to the center. The amplitude of the aerodynamic load response with yaw motion is smaller than with surge and pitch. The main effect of sway and roll motion is

to alter the direction of the wind perceived by rotor, whereas the impact on relative wind speed is limited. The aerodynamic response to motion in these directions is small and, due to the testing procedures we adopted, this is more evident for thrust than for torque. The aerodynamic thrust and torque are often introduced in control-oriented models as rotor-integral loads defined by means of the static thrust and torque coefficients or, when the model is linearized, with the derivatives of static coefficients with respect to rotor speed, wind speed, and pitch angle. We examined the response of such quasi-static aerodynamic model

to the harmonic oscillation of wind speed created by surge and pitch motion. The turbine loads measured in the experiments with surge motion are aligned to the model predictions, except for motion at wave frequency. Here, blade-level unsteady aerodynamics may occur, which is not accounted for by the quasi-static model. The model predictions are instead good for low-frequency motion. In case of pitch, the amplitude of thrust and torque oscillations with low-frequency motion is predicted with reasonable accuracy by the quasi-static model, but the phase is not. The difference in phase shift can be due to the inflow

conditions created pitch motion which is non-uniform across the rotor, or by other phenomena not considered in this analysis. Also with pitch, the quasi-static model is not predictive for wave-frequency motion.

The analysis of the aerodynamics-load response led to the following conclusions. The fact that the aerodynamic thrust and torque are largely influenced by surge and pitch motion, supports the idea of including these two degrees-of-freedom in coupled models of floating turbines. Reduced-order models with focus on the global dynamics of the system may instead

neglect motion in the other directions. Quasi-static aerodynamic models are predictive for low-frequency surge motion. The QST model is sensitive to the accuracy of the static coefficients from which it is derived. Discrepancies are seen in case of wave-frequency motion and pitch motion in general. Since the turbine response in the pitch direction is largely influenced

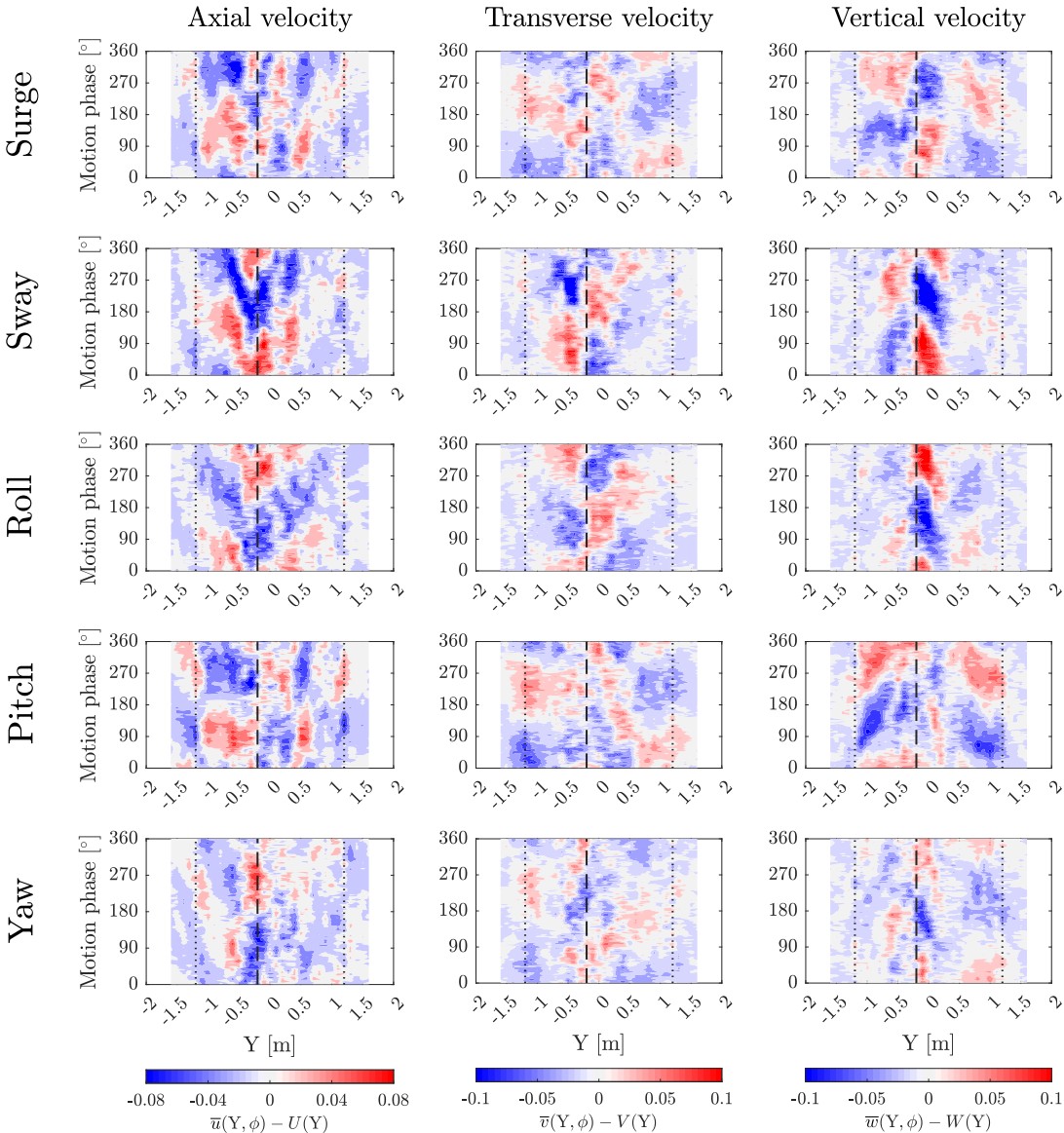

**Figure 9.** Space-time evolution of phase-averaged wake velocities ($\overline{u}(\mathrm{Y},\phi)$ axial, $\overline{v}(\mathrm{Y},\phi)$ transverse, $\overline{w}(\mathrm{Y},\phi)$ vertical, $\phi$ is the motion phase): variation with respect to the mean velocity in each point ($U(\mathrm{Y})$, $V(\mathrm{Y})$, $W(\mathrm{Y})$). The results are obtained for motion at the wave frequency, for WC2, at $\mathrm{X} = 2.3\mathrm{D}$. The phase-averaging is done based on 94 motion cycles. The vertical dotted lines mark the edge of the rotor and the dashed vertical line is in correspondence of the average position of the wake center.

by aerodynamic thrust force, and this is also the root cause of control issues in floating turbines, it is advised that future experimental efforts focus on pitch motion.

Wind speed in the turbine wake was measured at hub-height, 2.3D downwind the rotor, for the fixed turbine and imposing platform motion in five directions; the motion frequency was equal to the wave frequency at the Gran Canaria site (Mahfouz et al. (2021)). The experiment shows that the average axial-velocity in the wake in correspondence of the rotor disk is slightly lower with motion compared to the fixed case. In the motion conditions at hand, the wake recovery appears to be slower than for a bottom-fixed turbine. The turbulence kinetic energy in the wake is generally lower with motion than for a bottom-fixed

turbine, hence the wake is more stable. The lower flow mixing may explain the lower wake recovery. Phase-averaged results show that motion affects the three velocity components, which have marked oscillations at the same frequency of the imposed motion. The effect is stronger in WC2, so in high winds, compared to WC1. Motions in different directions affect the wake response in different ways. Surge, pitch, sway, and roll are responsible of periodic oscillations of the axial velocity across the entire wake width; additionally, pitch introduces oscillations in the vertical velocity, which suggests the wake moves in the X-Z

plane. Sway and roll have a strong effect on the transverse and vertical velocities, and this is more pronounced in the wake core region. With yaw motion the fluctuations of the axial velocity are confined to a narrow region in the center of the wake. No significant motion of the wake core is detected, so the velocity oscillation in the center of the wake that are seen with sway, roll, and yaw may be the result of the interaction between root vortices. Further research is needed to confirm this hypothesis and, in general, to explain the velocity fluctuations caused by platform motion. The oscillations in the wake velocity observed

for motion in different directions should be taken into account in studies about floating wind farms, because they represent an additional forcing for waked turbines. Moreover, the wake response induced by platform motion rises questions about if and how dynamic platform motion can be controlled and exploited to redirect the wake in space, and hence for wind farm control purposes. Concerning the wake of floating turbines, suggestion for future work is to carry out measurements with low-frequency motion conditions and measure the wake also in positions further downstream. Here, we considered a non-turbulent

inflow condition to highlight the effects of platform motion, but measurement with turbulence are advisable to understand the effect of inflow turbulence on wake mixing.

*Data availability.* The dataset is accessible upon request to the authors.

## Appendix A:  Test matrices

**Table A1.** Test matrix for cases with motion. $f_m$ is the frequency of motion, $f_r$ the rotor reduced-frequency, $A_m$ the amplitude, TSR is the tip-speed ratio and CP the collective pitch. Wake measurements were carried out for load cases where a $\times$ is present in the "Wake" column.

| Type | Wind speed [m/s] | $f_m$ [Hz] | $f_r$ [-] | $A_m$ [m | °] | Rotor speed [rpm] | TSR [-] | CP [°] | Wake |
|------|------|------|------|------|------|------|------|------|
| Surge | 2.95 | 0.350 | 0.280 | 0.060 | 210 | 9.0 | -3.0 | |
| Surge | 2.95 | 0.175 | 0.140 | 0.114 | 210 | 9.0 | -3.0 | |
| Surge | 2.95 | 3.175 | 2.540 | 0.006 | 210 | 9.0 | -3.0 | $\times$ |
| Surge | 5.00 | 0.350 | 0.168 | 0.114 | 216 | 5.4 | 8.5 | |
| Surge | 5.00 | 0.175 | 0.084 | 0.180 | 216 | 5.4 | 8.5 | |
| Surge | 5.00 | 3.175 | 1.524 | 0.013 | 216 | 5.4 | 8.5 | $\times$ |
| Sway | 2.95 | 0.350 | 0.700 | 0.038 | 210 | 9.0 | -3.0 | |
| Sway | 2.95 | 0.175 | 1.200 | 0.022 | 210 | 9.0 | -3.0 | |
| Sway | 2.95 | 3.175 | 2.540 | 0.011 | 210 | 9.0 | -3.0 | $\times$ |
| Sway | 5.00 | 0.350 | 0.420 | 0.064 | 216 | 5.4 | 8.5 | |
| Sway | 5.00 | 0.175 | 0.720 | 0.037 | 216 | 5.4 | 8.5 | |
| Sway | 5.00 | 3.175 | 1.524 | 0.018 | 216 | 5.4 | 8.5 | $\times$ |
| Roll | 2.95 | 0.700 | 0.560 | 1.900 | 210 | 9.0 | -3.0 | |
| Roll | 2.95 | 0.875 | 0.700 | 1.500 | 210 | 9.0 | -3.0 | |
| Roll | 2.95 | 3.175 | 2.540 | 0.400 | 210 | 9.0 | -3.0 | $\times$ |
| Roll | 5.00 | 0.700 | 0.336 | 3.000 | 216 | 5.4 | 8.5 | |
| Roll | 5.00 | 0.875 | 0.420 | 2.600 | 216 | 5.4 | 8.5 | |
| Roll | 5.00 | 3.175 | 1.524 | 0.700 | 216 | 5.4 | 8.5 | $\times$ |
| Pitch | 2.95 | 0.700 | 0.560 | 1.150 | 210 | 9.0 | -3.0 | |
| Pitch | 2.95 | 0.875 | 0.700 | 1.000 | 210 | 9.0 | -3.0 | |
| Pitch | 2.95 | 3.175 | 2.540 | 0.250 | 210 | 9.0 | -3.0 | $\times$ |
| Pitch | 5.00 | 0.700 | 0.336 | 2.760 | 216 | 5.4 | 8.5 | |
| Pitch | 5.00 | 0.875 | 0.420 | 2.200 | 216 | 5.4 | 8.5 | |
| Pitch | 5.00 | 3.175 | 1.524 | 0.600 | 216 | 5.4 | 8.5 | $\times$ |
| Yaw | 2.95 | 2.625 | 2.100 | 0.400 | 210 | 9.0 | -3.0 | |
| Yaw | 2.95 | 0.375 | 0.300 | 2.600 | 210 | 9.0 | -3.0 | |
| Yaw | 2.95 | 3.175 | 2.540 | 0.300 | 210 | 9.0 | -3.0 | $\times$ |
| Yaw | 5.00 | 2.625 | 1.260 | 0.730 | 216 | 5.4 | 8.5 | |
| Yaw | 5.00 | 0.375 | 0.180 | 3.000 | 216 | 5.4 | 8.5 | |
| Yaw | 5.00 | 3.175 | 1.524 | 0.600 | 216 | 5.4 | 8.5 | $\times$ |

## Appendix B: The wind tunnel

This appendix provides additional details about the wind tunnel facility and the instruments utilized for force and wake measurements. The wind tunnel of Politecnico di Milano is shown in Fig. B1. Testing was conducted in the atmospheric boundary layer section which is placed in the return duct of the wind tunnel. The turbine model was placed on top of a 6-DOFs robotic platform (not shown in the figure), and the rotor plane was at about 27.5 m from the test section inlet. The wind velocity in the turbine wake was measured with two three-components hot-wire probes (*Dantec 55R91*) mounted on a traversing system

at 100 mm of Y-distance one from the other and moved in the Y direction from -1.6 m to 1.6 m at constant Z = 2.15 m (dashed line in Fig. B1). The conditioning system for the hot-wire probes is formed by a *Dantec StreamLine* chassis and *Dantec CTA 90C10* anemometric modules. The calibration procedure and the data acquisition software were developed in house and are confidential. The bandwidth of the measurement system is about 10 kHz. The 6-components tower-nacelle interface forces were measured with a *ATI Mini45* with *SI-580-20* calibration.

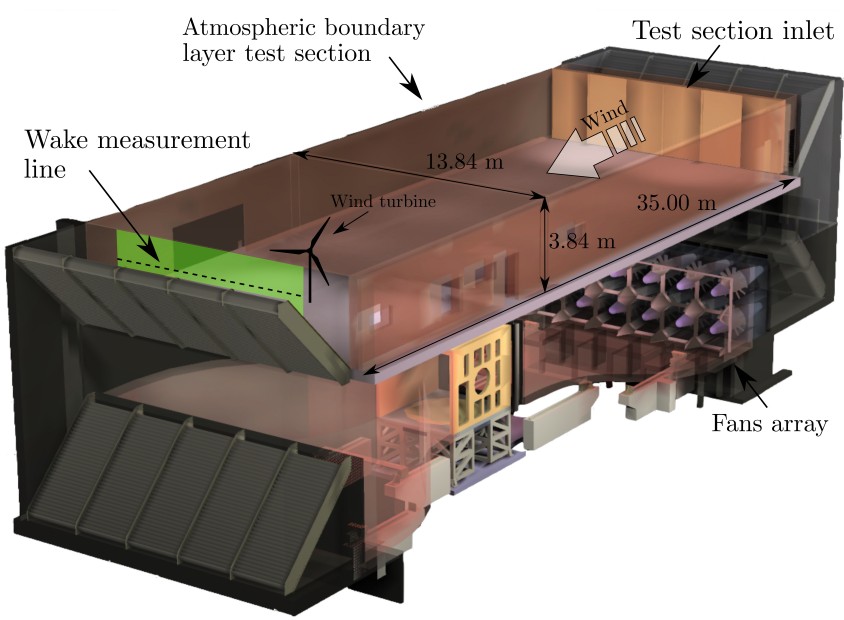

**Figure B1.** Sketch of the wind tunnel with the main dimensions of the atmospheric boundary layer test section where the experiments were carried out, and position of the wake measurement line.

*Author contributions.* AFo and AFa designed the wind tunnel experiments, which were carried out with the help of SDC. Data were processed by AFo. MB and AFa supervised the work and the project in a larger perspective. All coauthors thoroughly reviewed the article.

*Competing interests.* The authors declare that they have no conflict of interest.

*Acknowledgements.* This research, including the open-access publication, has been supported by Horizon 2020 project COREWIND (grant no. 815083).

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
