# Peer review of "Wind tunnel investigation of the aerodynamic response of two 15 MW floating wind turbines"

_Wind Energy Science, 2022_

## Referee Comment (RC1)

Review of WES-2022-36

**Wind tunnel investigation of the aerodynamic response of two 15 MW floating wind turbines – Alessandro Fontanella et al.**

The work investigates the aerodynamic response of a scaled 15MW rotor. The rotor is excited in Surge, Pitch, Sway, Roll and Yaw at the natural frequencies of two floaters that were designed for this rotor and at wave frequency. The article is well written, clearly structured and results appear credible and solid. The topic is scientifically relevant. I have some minor remarks for the authors:

As a general remark, although the wind tunnel dimensions are stated in the article, it would be useful to have a sketch of the wind tunnel, including its dimensions, shape, and the position of the wind turbine within it. This could go into an appendix.

L173-175: please rephrase, it is not completely clear which "last two conditions" the authors are referring to

L176-179: Is the response expected to be non-linear with respect to amplitude? The reason for testing two amplitudes should be explained more clearly

Fig.2: I understand that Heave was not tested due to limitation in wind tunnel height, however makers are present in both left and right figures which is confusing. Please either remove the heave frequency, or indicate it with different shapes/colors and add an appropriate statement to the figure legend.

L198-200. This statement is relative to all DOFs presented in page 8? What do the authors mean for "limit case"?

Figure 3: The values of Cp in the map on the right are confusing if compared to Table 3. Are these the values for the scaled model? According to table 3 Cp should be 0.35 in the "below" rated operating condition, while a value between 0.45 and 0.5 can be seen in figure 3.

L 248-250: Is it possible to estimate the effect of inertial torque? Significant variation in torque is noted in fig. 5 for roll, not matched by thrust variations. Is this caused by aerodynamic unsteadiness or does it result from inertial torque that could not be removed?

Section 5: I would suggest "wake measurement results" as an alternative section title

Figure 7: include oscillation frequency of measurements in the description

L 338-340: This statement can be confusing if read together with lines 349-351, where the contrary is suggested. I would suggest to rephrase in order to more clearly specify that the former statement is referred to low-frequency motion while the latter to high-frequency.

L 360-365: It is not clear if the meandering of the wake is noticed in above rated conditions only, or if it is noted below rated too. Conclusion suggest that this is only noted above rated but please clarify. Also, the oscillations in wake center may be interesting to investigate in more detail.

L385-387: If I interpret correctly, the quasi-static model performs well for surge, but does not predict the correct phase shift for pitch. It would be nice to elaborate on this with eq.1 and fig. 2 in mind. Perhaps rotor-level unsteadiness is more influent for pitch than for surge?

Minor text editing, for example: L 339: "helps to promote" instead of "helps promoting", L 372: "forced to move" not "forced moving"

---

## Author Comment (AC1)

Politecnico di Milano
Department of Mechanical Engineering
Via La Masa 1, 20156, Milan
Italy

Wind Energy Science Discussion

Date: June 06, 2022
Subject: WES-2022-36 Final Response

Dear Referees,

We would like to thank you for the accurate review and the valuable feedbacks. Your suggestions focused our attention on aspects that we didn't consider, and we believe this will improve the quality and impact of this work.

We are sure you devoted a good amount of time in the review of our article, and we are grateful for that.

The article has been revised following your suggestions. The attached document provides a detailed answer to the comments you made in the Interactive Discussion.

On behalf of all Authors,
yours sincerely,

Alessandro Fontanella

Attached documents:
- Response to Anonymous Referee #1
- Response to Anonymous Referee #2

**Response to Anonymous Referee #1**

Dear Referee,

Thank you for taking the time to review our manuscript and for the valuable comments you made. Below are our answers.

| | |
|---|---|
| **RC1.1** | As a general remark, although the wind tunnel dimensions are stated in the article, it would be useful to have a sketch of the wind tunnel, including its dimensions, shape, and the position of the wind turbine within it. This could go into an appendix. |
| **AC1.1** | As you suggested, we added a new appendix at the end of the article, where we included a sketch of the wind tunnel with its main dimension and the position of the wind turbine model. Moreover, we used the new appendix to provide the reader additional information about the hot-wire probes and the load-cell as asked by Referee #2. |

| | |
|---|---|
| **RC1.2** | L173-175: please rephrase, it is not completely clear which "last two conditions" the authors are referring to |
| **AC1.2** | We changed the sentences: |
| | "*Motion with frequency of the WindCrete yaw mode and at the wave frequency may result in some unsteadiness for the blade section with r/R < 0.5. For the last two conditions, blade-level and rotor-level unsteady aerodynamics may occur together, whereas only rotor-level unsteadiness is expected at the other motion frequencies.*" |
| | in |
| | "*Motion with frequency of the WindCrete yaw mode and at the wave frequency may result in some unsteadiness for the blade sections with r/R < 0.5, and so in blade-level and rotor-level unsteady aerodynamics occurring together. Only rotor-level unsteadiness is expected at the other frequencies of motion.*" |
| | In this way, we think that blade-level and rotor-level unsteadiness occur together for motion with the frequency of the WindCrete yaw mode and with the wave frequency. |

| | |
|---|---|
| **RC1.3** | L176-179: Is the response expected to be non-linear with respect to amplitude? The reason for testing two amplitudes should be explained more clearly |
| **AC1.3** | Thank you for having mentioned this. We added a few sentences in the section about the selection of the amplitude of motion to explain why we considered two values of amplitude for each frequency: |
| | "*The aerodynamic rotor loads are expected to be linearly proportional to the rotor apparent wind, as it is found by Fontanella et al. (2021) and Mancini et al. (2020) for the case of low-frequency surge motion. In this test campaign the three frequencies fm were tested with two values of amplitude for any platform DOF; in the case of motion in surge, pitch and yaw directions, which creates an apparent wind, this is done to confirm the linearity of aerodynamic loads with respect to the apparent wind (i.e. to motion amplitude when the frequency is*" |

*fixed). Two amplitudes were tested also in the case of sway and roll motion to cover a wider range of conditions."*

| **RC1.4** | Fig.2: I understand that Heave was not tested due to limitation in wind tunnel height, however makers are present in both left and right figures which is confusing. Please either remove the heave frequency, or indicate it with different shapes/colors and add an appropriate statement to the figure legend. |
|---|---|
| **AC1.4** | This is correct, heave was not tested. We decided to remove the markers in correspondence of the heave frequency from the two plots. |

| **RC1.5** | L198-200. This statement is relative to all DOFs presented in page 8? What do the authors mean for "limit case"? |
|---|---|
| **AC1.5** | We realized it is difficult to correlate the amplitude of harmonic motion to the response of a floating to stochastic wind and waves. Some options are available: for example, one can take the amplitude at the frequency of motion of the complex spectrum of the platform response, or use the maximum of the spectrum, or other metrics like the standard deviation of the motion time series. For this reason, we preferred to change the text in:

"*The values of amplitude and frequency of the surge and the pitch motion are comparable to those considered by Ramos-García et al. (2021) to investigate the aerodynamic response of the IEA 15 MW mounted on the WindCrete floater by means of two different aerodynamic solvers."*

The full-scale values of motion amplitude are recalled in the bulleted list at lines 204-216 or can be computed from the table in the appendix A. We leave to the reader the comparison with his metric of interest for the full scale turbine. |

| **RC1.6** | Figure 3: The values of Cp in the map on the right are confusing if compared to Table 3. Are these the values for the scaled model? According to table 3 Cp should be 0.35 in the "below" rated operating condition, while a value between 0.45 and 0.5 can be seen in figure 3. |
|---|---|
| **AC1.6** | You are right and what you write about the Cp values is correct. We already faced this issue in wind tunnel testing of other rotors, and it deserves to be explained better. We modified Figure 3 adding the values of Table 3 close to the corresponding markers and improving the contrast of contour lines. |

[Figure]

**Figure 3.** Map of the power and thrust coefficients of the turbine model measured in steady wind with $U_\infty = 4$ m/s. The × marks correspond to the turbine operating conditions of Table 4, and the coefficient values obtained in tests with fixed turbine are reported in brackets.

Then, we added this text to the Figure 3 comment:

"*The values of power and thrust coefficient obtained from interpolation of the maps for the WC2 are similar to those measured in the experiment with WC2 wind. The same is not true for WC1, and values obtained from the maps are higher than those measured in the experiment at 3 m/s. The aerodynamic behavior of the blade is sensitive to the Reynolds number, which is lower in WC1 than with the 4 m/s for which the maps were obtained. The lower Reynolds results in a slightly lower lift force and higher drag, and so in a lower power coefficient and slightly lower thrust coefficient.*"

This discrepancy due to Reynolds is an additional uncertainty for the QST model, which is obtained based on the Cp and Ct maps measured at 4 m/s. We commented this in Section 4.2.1 (lines 333-337):

"*Figure 3 shows the power coefficient and, to a lesser extent, the torque coefficient depend on the wind speed, and values measured at 3 m/s are not coincident with those predicted by the CP and CT maps that were obtained at 4 m/s. In Eq. 16-17 we assume the derivatives of the CT and CQ do not depend on the wind speed. This assumption is a source of uncertainty for the QST model, for torque more than for thrust, because the aerodynamic torque is more sensitive to Reynolds number.*"

| RC1.7 | L 248-250: Is it possible to estimate the effect of inertial torque? Significant variation in torque is noted in fig. 5 for roll, not matched by thrust variations. Is this caused by aerodynamic unsteadiness or does it result from inertial torque that could not be removed? |
|---|---|
| AC1.7 | The dynamic torque we see with motion in the sway and roll direction is mostly due to mechanical inertia which is not removed in the force post processing. As you said, it is very unlikely to have large aerodynamic torque oscillations with no thrust oscillations. The reason why part of the mechanical inertia is not removed is the test procedure we used. Indeed, it wasn't possible to keep the rotor still in tests without wind because of the compliance of the turbine controller. In tests without wind the drivetrain is free to rotate differently than in tests with wind where the rotor speed is fixed. This flexibility is the reason of the dynamic torque we see in the results. |

We added this text at the end of Sect. 4.2 to clarify the results we got and why it is difficult to remove entirely the mechanical inertia:

"*The amplitude is about linearly proportional to the frequency of motion and the phase shift is around −π; the torque maximum is when the acceleration due to platform motion is directed as the blade peripheral speed. The large amplitude of torque variations, which is not matched by thrust, and their phase shift suggest the torque response is mostly due to mechanical inertia which could not be removed by the force post-processing. To completely remove the mechanical inertia of the rotor it is necessary to have the rotor perfectly still in tests without wind, but this proved impractical due to the compliance of the turbine controller which allowed small oscillations of the rotor (less than 10° of amplitude) in tests without wind. In principles, the mechanical inertia of the rotor could be estimated also with tests with spinning rotor, but these must be run in the void, and it is unfeasible given the size of the turbine model.*"

Given the difficulties in removing the mechanical inertia, we think the results are still valuable:

*"Given the test procedure adopted in this campaign it is not possible to isolate the oscillations of aerodynamic torque due to unsteady wind only, however we can reasonably say these are small enough to be masked by the uncertainties associated with the testing."*

We also included these two sentences in the conclusion:

*"The main effect of sway and roll motion is to alter the direction of the wind perceived by rotor, whereas the impact on relative wind speed is limited. The aerodynamic response to motion in these directions is small and, due to the testing procedures we adopted, this is more evident for thrust than for torque."*

| | |
|---|---|
| **RC1.8** | Section 5: I would suggest "wake measurement results" as an alternative section title |
| **AC1.8** | The title of Section 5 has been changed according to your suggestion. |

| | |
|---|---|
| **RC1.9** | Figure 7: include oscillation frequency of measurements in the description |
| **AC1.9** | The results of Fig. 7 and Fig. 8 are obtained with motions of frequency equal to the wave frequency. We stated this in the caption of the figures: |
| | *"Average wake deficit at X = 2.3D for the fixed case and with different motions at the wave frequency"* |
| | and |
| | *"Turbulence kinetic energy at X = 2.3D for the fixed case and with different motions at the wave frequency"* |

| | |
|---|---|
| **RC1.10** | L 338-340: This statement can be confusing if read together with lines 349-351, where the contrary is suggested. I would suggest to rephrase in order to more clearly specify that the former statement is referred to low-frequency motion while the latter to high-frequency. |
| **AC1.10** | Yes, you are right, we didn't realize this. We replaced the text: |
| | *"Ramos-García et al. (2021) and Fu et al. (2019) observed the wake recovery for a pitching turbine is different than in the bottom-fixed case: flow mixing is higher for a floating turbine because increased turbulence due to motion helps promoting a faster break down of the strong vortex structure"* |
| | with |
| | *"Ramos-García et al. (2021) and Fu et al. (2019) observed the wake recovery for a FOWT which undergoes to pitch motion with a frequency close to the platform pitch mode is different than in the bottom-fixed case. In this case flow mixing is higher for a floating turbine because increased turbulence due to low-frequency motion promotes a faster break down of the strong vortex structures"* |

| | |
|---|---|
| **RC1.11** | L 360-365: It is not clear if the meandering of the wake is noticed in above rated conditions only, or if it is noted below rated too. Conclusion suggest that this is only noted above rated but please clarify. Also, the oscillations in wake center may be interesting to investigate in more detail. |
| **AC1.11** | The Referee #2 asked also to extend the analysis of wake measurements to the transvers and vertical velocity components and to other positions than the wake |

center. The new results and the corrections we made to the text are discussed in AC2.9.

| | |
|---|---|
| **RC1.12** | L385-387: If I interpret correctly, the quasi-static model performs well for surge, but does not predict the correct phase shift for pitch. It would be nice to elaborate on this with eq.1 and fig. 2 in mind. Perhaps rotor-level unsteadiness is more influent for pitch than for surge? |
| **AC1.12** | Thank you for this suggestion. We added these comments to the results of Section 4.2.1:

"*In case of pitch, unlike what happens with the surge, the phase of the aerodynamic loads measured in the experiment is never $-n/2$. As it is shown in Fig. 2, blade-level unsteadiness can be the cause of the different phase shift only in the case of pitch motion with wave frequency but is not a valid explanation for cases with low-frequency motion. The rotor reduced-frequency is higher for low-frequency pitch motion compared to surge ($2\times$ for the WindCrete and $5\times$ for the Activefloat) and this may give some rotor-level unsteadiness. Another reason for the different phase behavior may be the skewed inflow created by pitch motion. Surge and pitch are equivalent in terms of the apparent wind speed at hub height, but pitch motion produces a velocity gradient across the rotor height, with the upper portion feeling a higher wind speed than the lower one. This is not accounted for by the QST model, which models the rotor as a single point coincident with the hub.*" |

| | |
|---|---|
| **RC1.13** | Minor text editing, for example: L 339: "helps to promote" instead of "helps promoting", L372: "forced to move" not "forced moving" |
| **AC1.13** | Checked. |

**Response to Anonymous Referee #2**

Dear Referee,

Thank you for the extended and accurate feedback. We see you put a significant effort in reviewing our article, and we are grateful for this.

We think your comments helped us improving the quality of this paper and of our research in general, pushing us to extend our analyses beyond what we did for the first draft.

You can find below the response to your comments.

**Major comments**

| **RC2.1** | The experimental set-up is not comprehensive enough. Give technical details on the hot wire anemometry that you use (probe, anemometers, cut-off frequency of the probe, calibration procedure, etc …the brand? ). The same for the load cells. |
|---|---|
| **AC2.1** | Referee #1 suggested to add a new appendix with a sketch of the wind tunnel and its dimensions. We used this appendix to provide the reader some technical details about the instruments we used for force and wake measurements. |
| | *"The wind velocity in the turbine wake was measured with two three-components hot-wire probes (Dantec 55R91) mounted on a traversing system at 100 mm of Y-distance one from the other and moved in the Y direction from -1.6 m to 1.6 m at constant Z = 2.15 m (dashed line in Fig. B1). The conditioning system for the hot-wire probes is formed by a Dantec StreamLine chassis and Dantec CTA 90C10 495 anemometric modules. The bandwidth of the measurement system after the calibration procedure, which is confidential, is about 10 kHz. The 6-components tower-nacelle interface forces were measured with a ATI Mini45 with SI-580-20 calibration."* |

| **RC2.2** | Page 2, lines 149-150: More explanations are needed on the frequency choices. Give the full and model scale values of the natural frequencies for ActiveFLoat and WindCrete. Explain why you choose to add a configuration with the frequency of the wave spectrum peak of a specific site. Do we really expect to see the floaters moving at this frequency? |
|---|---|
| **AC2.2** | As you suggested, we added a table with the full-scale and model scale values of the natural frequencies and wave frequency for the Activefloat and WindCrete floating systems. |

**Table 3.** Full-scale and model scale values of the natural frequencies and wave frequency for the Activefloat and WindCrete. Full-scale values are taken from (Mahfouz et al. (2021)).

| FOWT | Surge | Sway | Roll | Pitch | Yaw | Wave |
|---|---|---|---|---|---|---|
| WindCrete full-scale [Hz] | 0.012 | 0.012 | 0.024 | 0.024 | 0.092 | 0.111 |
| Activefloat full-scale [Hz] | 0.006 | 0.006 | 0.031 | 0.031 | 0.012 | 0.111 |
| WindCrete model scale [Hz] | 0.350 | 0.350 | 0.700 | 0.700 | 2.625 | 3.175 |
| Activefloat model scale [Hz] | 0.175 | 0.175 | 0.875 | 0.875 | 0.375 | 3.175 |

Concerning the motion at wave frequency:

"*In general, the motion of a FOWT is due to wind and wave excitation. In mild waves, the platform motion is driven by wind and second-order hydrodynamic loads, at it is mainly at the natural frequencies of the rigid-body modes; the amplitude of motion at wave frequency depends on the strength of waves and on the wave direction.*"

| | |
|---|---|
| **RC2.3** | On general, once you have defined the reduced frequencies give the values in reduced frequencies only (in body text but also in figure captions and legend). |
| **AC2.3** | Thank you for this suggestion. We think it makes it easier to read the results and it favors comparison with results of other works. We used the reduced frequency in the text, and we added a new column for it in Table A1. |

| | |
|---|---|
| **RC2.4** | Page 2, lines 166-168: explain where does this threshold comes from. Explain what it means and what is the physics behind. |
| **AC2.4** | We revised the text and the Eq. (2), and we added a few comments to explain the physical meaning of the threshold blade-reduced frequency. |

"At the same time, blade-level unsteadiness may occur, as predicted for example by Theodorsen theory, when the blade reduced-frequency $f_c$ is high. The blade-reduced frequency $f_c$ is defined as:

$$f_c = \frac{\pi f_m c}{V}$$

where $c$ is the chord of a blade section and $V$ is the velocity it perceives. When $f_c$ is small the circulatory contributions to airfoil lift from Theodorsen's theory dominate; when $f_c$ is high, the apparent mass contributions, which arise from flow acceleration effects, begins to dominate, and flow unsteadiness is expected to take place. As explained by Sebastian and Lackner (2013), unsteadiness typically occurs for $f_c \geq 0.05$, that is when the apparent mass effects due to local flow acceleration become meaningful."

| | |
|---|---|
| **RC2.5** | Definition of load cases: the present experiments are performed for constant blade pitch and rotor speed. The terms "Above-rated" and "below rated" refer to different control strategies (fixed rotor speed OR fixed blade pitch) that you do not respect here. Consequently, you are simply testing to different steady operating points. So do not use the terminology AR and BR because it could confuse the reader |
| **AC2.5** | We used below rated and above rated to identify the two functioning conditions because one is for a wind speed below the turbine rated speed, and one for a wind speed above it. However, we recognize that below-rated and above-rated are often used in control-related topics and can confuse the reader. We replaced them with "wind condition 1" (WC1) and "wind condition 2" (WC2). |

| | |
|---|---|
| **RC2.6** | Figure 5: if I have well understood, the plots are built with sine functions using the Thrust or torque amplitudes and phases obtained by the experiments. Wouldn't it be more synthetic to present a 2D-plot of the amplitude or phase versus the motion amplitude (x axis) and frequency (y axis)? |

**AC2.6** Yes, what you write is correct, and we agree it is difficult to make quantitative comments with the plot we showed in the draft. We replaced it with a new one, that shows the amplitude and phase-shift with respect to motion as a function of the rotor-reduced frequency. We decide to use the same y-axis limits for all motion directions to highlight the motion conditions that have the largest influence on the rotor force response.

[Figure]

**Figure 5.** Amplitude and phase shift of the dynamic thrust force ($\overline{\Delta F_x}$, $\angle\Delta F_x$), and amplitude and phase shift of the dynamic torque ($\overline{\Delta M_x}$, $\angle\Delta M_x$) as function of reduced frequency $f_r$. Reversed-triangle markers correspond to results for the wind condition 1 (WC1), triangle markers to results for the wind condition 2 (WC2), and colors identify different values of the motion amplitude $A_m$.

**RC2.7** Equations 14 and 15 : Explain these formula. It is not explained how they are established.

**AC2.7** We added a couple of equations more to explain how the equations you mentioned are established. The new text is:

"*The sensitivities $K_{U,T}$ and $K_{U,Q}$ are computed from the thrust and torque coefficients respectively. The expressions of $K_{U,T}$ as function of $C_T$ and of $K_{U,Q}$ as function of $C_Q$ are derived computing the derivative of Eq. 4-5 with respect to the wind speed $U$.*

$$K_{U,T} = \left.\frac{\partial F_x}{\partial U}\right|_0 = \left(\rho C_T A_r U\right)_0 + \left(\frac{1}{2}\rho A_r U^2 \frac{\partial C_T}{\partial U}\right)_0,$$

$$K_{U,Q} = \left.\frac{\partial M_x}{\partial U}\right|_0 = \left(\rho C_Q A_r R U\right)_0 + \left(\frac{1}{2}\rho A_r R U^2 \frac{\partial C_Q}{\partial U}\right)_0.$$

*In a more compact form, Eq. 14-15 are*

$$K_{U,T} = \frac{\partial F_x}{\partial U}\bigg|_0 = \frac{F_{x,0}}{U_\infty}\left(2 - \frac{\partial C_T}{\partial \lambda}\bigg|_0 \frac{\lambda_0}{C_{T,0}}\right),$$

$$K_{U,Q} = \frac{\partial M_x}{\partial U}\bigg|_0 = \frac{M_{x,0}}{U_\infty}\left(2 - \frac{\partial C_Q}{\partial \lambda}\bigg|_0 \frac{\lambda_0}{C_{Q,0}}\right),$$

*where $\lambda$ is the TSR."*

| | |
|---|---|
| **RC2.8** | Figure 6 : the linear scale used for the x axis is not appropriate to the chosen configurations. Having the points for the highest frequency on the plots makes necessary to zoom out and makes impossible to see whether the points are in agreement with the quasi-static model. Please find a better way to plot these results to make visible the agreement between model and experiments. |
| **AC2.8** | Sure, you are right, it wasn't possible to draw any conclusion about the low-frequency points. We added an insert to the figure, to zoom on the low-frequency points. |

[Figure]

| | |
|---|---|
| **RC2.9** | Time series of velocity within the wake (figure 9) : It is chosen to study the influence of the motion only on the axial velocity time series at hub height, so in the wake centre. One can though expect to better observe the influence of the motion on other velocity components ( for instance on the lateral velocity component for the sway or yaw) or on other locations (for instance, the wake edges, in the shear layer, where the meandering signature is often easier to observe). Therefore, it is essential to extend the study to other velocity components and space locations before interpreting these results. |
| **AC2.9** | Thank you for this comment. We initially focused on the velocity in correspondence of the wake center because we thought it would be easier to interpret the results. However, as you said, one gets a very limited view of what happens in the wake when he focusses on just one point. We embraced your |

suggestion, and we extended the analysis to the phase-averaged time series of any location across the wake. What we got is shown in the new figure 9.

[Figure]

**Figure 9.** The time series show the average variation in one motion cycle of the wind velocity across the turbine wake for WC2, at X = 2.3D. The time series are the phase-average of 94 motion cycles and are deprived of the mean value. The vertical dotted lines mark the edge of the rotor and the dashed vertical line is in correspondence of the average position of the wake center.

The new comment to Fig. 9 reads:

"*Figure 9 shows the phase-averaged time series of the variation of the three wind speed components in the turbine wake for WC2, at X = 2.3D and different Y positions. Results for WC1 are omitted because, the amplitude of velocity oscillations is larger in WC2 compared to WC1. In WC1 the amplitude of axial-velocity fluctuations associated with motion is small, less 1% of $U_\infty$, and similar in magnitude to the turbulence in the wake of the fixed turbine. Moreover, it is easier to interpret results for WC2 compared to WC1 because in WC2 the wake symmetry with respect to the X axis is better. The effect of motion in the surge direction is visible in the axial and vertical velocity components. In the first half of the motion cycle, the axial velocity is increased across the entire wake, and it is decreased in the second half; the variation is stronger on the left of the average wake center position compared to the right. The variation in the wake velocity is the effect of the perturbation introduced by the rotor when it moves, that is propagated downstream. Thus, the phase with respect to motion of the velocity*

*variation depends on the propagation speed and the distance from rotor. For the vertical component, the peak velocity is reached at different times by sections in different Y positions; the peak is reached first on the left side and then it propagates towards the right. Motion in the sway and roll directions has a clear effect on all three velocity components. The axial velocity is increased periodically across the entire wake, but, differently than with surge, the peak in the outer sections is delayed moving from the wake center to the wake edges. We applied the tracking method described by Coudou et al. (2018) to the axial velocity data to detect the position of the wake center, and this does not change significantly in one sway/roll cycle. If the wake core moves, the motion is smaller than the spatial resolution of wake data (i.e., 100 mm, or 10 m full-scale). Sway and roll excite in a significant manner the transverse and vertical velocity; the response is stronger in the central portion of the wake and negligible on the edges; the maxima and minima travel periodically from left to right. These findings suggests that sway and roll motion affect the rotation of the central portion of the wake. The effect of pitch motion on the axial velocity is similar to surge, but slightly more pronounced. Differently than with surge, pitch motion has a strong effect on the vertical velocity, which is periodically increased and decreased across the wake width. This supports the idea that oscillating pitch moves the wake up and down in the X-Z plane. The response to oscillating yaw is visible mostly in the axial velocity and in a narrow region in the center of the wake. Here the velocity is increased and decreased periodically. There are no visible effects on the other two components. The wake-center tracking algorithm does not detect any motion of the wake center also in the case of yaw motion. From these results, it seems that high-frequency yaw oscillations like those caused by waves do not move the wake center in a significant way, contrary to the case of static (or quasi-static) yaw, which is often used for redirecting the wake laterally (Meyers et al. (2022))".*

And in the conclusion:

"*Phase-averaged results show that motion affects the three velocity components, which have marked oscillations at the same frequency of the imposed motion. The effect is stronger in WC2, so in high winds, compared to WC1. Motions in different directions affect the wake response in different ways. Surge, pitch, sway, and roll are responsible of periodic oscillations of the axial velocity across the entire wake width; additionally, pitch introduces oscillations in the vertical velocity, which suggests the wake moves in the X-Z plane. Sway and roll have a strong effect on the transverse and vertical velocities, thus on the rotation of the wake core. With yaw motion the fluctuations of the axial velocity are confined to a narrow region in the center of the wake.*"

| | |
|---|---|
| **RC2.10** | Figure 9 the figure iS big, whereas it does contain limited information. Could it be replaced by table with the values of standard deviations or the amplitude of the phase averaged values? |
| **AC2.10** | We replaced Fig. 9 with a new one, as discussed in AC2.9. |

| | |
|---|---|
| **RC2.11** | Conclusion, Page 19, lines 386_388 : "The difference in phase shift can be due to the inflow conditions created pitch motion which is non-uniform across the rotor, or by other phenomena not considered in this analysis.". This statement appears for the first time in the conclusion and is not mentioned in the previous discussion related to these results. Additionally, does it mean that the inflow is not uniform enough to ensure the symmetry of the flow? |

**AC2.11** We added this statement in the Section 4.2.1, and we also tried to explain the effect of pitch motion on the inflow in a better way.

*"In case of pitch, unlike what happens with the surge, the phase of the aerodynamic loads measured in the experiment is never $-\pi/2$. […] Another reason for the different phase behavior may be the skewed inflow created by pitch motion. Surge and pitch are equivalent in terms of the apparent wind speed at hub height, but pitch motion produces a velocity gradient across the rotor height, with the upper portion feeling a higher wind speed than the lower one. This is not accounted for by the QST model, which models the rotor as a single point coincident with the hub."*

**Minor comments**

**RC2.12** Choose between "wave-peak frequency" and "wave frequency"

**AC2.12** We decided to use wave frequency.

**RC2.13** Line 18 : farther

**AC2.13** Checked.

**RC2.14** Line 21 : floating wind farms

**AC2.14** Checked.

**RC2.15** Line 66 : We

**AC2.15** Checked.

**RC2.16** Lines 72-74 : "Past test campaigns at Politecnico di Milano focused on the response to low-frequency motion where the movement of the system is large because of resonant excitation." : why is this information important in the present paper?

**AC2.16** Indeed, it's not that important. Our goal with this sentence was to introduce the motion conditions of this experiment. We replaced

*"Past test campaigns at Politecnico di Milano focused on the response to low-frequency motion where the movement of the system is large because of resonant excitation. Large motion is also expected in the wave frequency range, which is addressed in the present campaign."*

with

*"The motion of a FOWT is large in the low-frequency range, where resonant excitation occurs, and in the wave-frequency range, where hydrodynamic loading associated with waves is large. The effect of motion on the turbine aerodynamic response should be more pronounced at these frequencies, which are covered by the motion conditions of the experiment."*

| **RC2.17** | Lines 104-105 : "Graphs of the results section are made in accordance with the recommendations of Stoelzle and Stein (2021) to improve data perception." : why is this information important here? What do they do as special? |
|---|---|
| **AC2.17** | We liked the paper of Stoelzle and Stein (2021) and we think it really helped us improving the quality of our figures. Citing the paper is a way of acknowledging the authors for their work, and it can be helpful for interested readers. We added this text: |
| | *"Graphs of the results section are made in accordance with the recommendations of Stoelzle and Stein (2021) which should make plots easier to read; information in line graphs is coded with line type, perceptually uniform color maps are used for 2D plots, and the red-grey-blue color map is used to underline data direction in 2D plots with zero midpoint."* |

| **RC2.18** | Line 113 : the turbine has individual blade pitch control but this functionality is not used in the present paper : collective pitch control. |
|---|---|
| **AC2.18** | We agree it may be misleading, and we replaced "*individual blade-pitch control*" with "*collective blade-pitch control*". |

| **RC2.19** | Line 129 : two wind turbines |
|---|---|
| **AC2.19** | The sentence has been changed in "*The experiment considered two functioning conditions for the wind turbine, reported in Table 2*". |

| **RC2.20** | Table 2 : do not use CP but the greek symbol beta, as usual for blade pitch and define the reference of this blade pitch. beta =0° when…? |
|---|---|
| **AC2.20** | We replaced CP with $\beta$. We recognized it may be difficult to understand why the pitch angle in wind condition 1 is lower than zero, and we added some comments about this at the beginning of Sect. 3: |
| | *"The experiment considered two functioning conditions for the wind turbine, reported in Table 2. In one, the wind turbine is operated at the blade pitch and tip-speed ratio (TSR) that give the maximum power coefficient. A fine-trim search shows the maximum power coefficient is attained for TSR = 9.0 and $\beta$ = −3.5°, and these values were selected for WC1. The rotor was designed to have its maximum efficiency at TSR = 9.0 and $\beta$ = 0° as the IEA 15 MW, and the fact this occurs for a lower $\beta$ may be due to errors in the blade mounting. In WC2 the rotor speed is equal to its rated value, and the blade pitch is trimmed to reduce the rotor efficiency. The TSR is equal to the steady-state value for the IEA 15 MW at the corresponding wind speed, and $\beta$ is adjusted to match the power coefficient of the reference turbine. Also in WC2 the pitch offset with respect to the IEA 15 MW is found to be -3.5°."* |

| **RC2.21** | Lines 160-161 : "Results of different studies about the aerodynamic wind turbines are presented as function of fr in the article of Ferreira et al. (2021)." : what are the takeaway messages of this article? |
|---|---|
| **AC2.21** | We cited the article because also Ferreira uses the rotor-reduced frequency to compare results of different studies. Apart from this, we think it is difficult to |

compare the conclusions of his article to the results of this one, and we preferred to remove the sentence:

"*Results of different studies about the aerodynamic wind turbines are presented as function of $f_r$ in the article of Ferreira et al. (2021).*"
* * *
**RC2.22** Line 174: " For the last two conditions…" : which ones exactly?

**AC2.22** With "the last to conditions" we intended yaw motion with the frequency of the yaw mode of the WindCrete and motion with wave frequency. We agree with you it wasn't clear and we replaced the old text with:

"*The blade aerodynamic response is quasi-steady for motion at the natural frequencies of the two floating turbines. Motion with frequency of the WindCrete yaw mode and at the wave frequency may result in some unsteadiness for the blade sections with r/R < 0.5, and so in blade-level and rotor-level unsteady aerodynamics occurring together.*"
* * *
**RC2.23** Line 179: were instead of where

**AC2.23** Checked.
* * *
**RC2.24** Figure 2 : indicate "f_r" on the colorbar legend (you use 2 different reduced frequencies in the paper, so it could be confusing). Difficult to see the ActiveFloat symbols

**AC2.24** We modified the left plot of Fig. 2 using $f_r$ in the colorbar legend and we replaced symbols with vertical lines as in the right plot.

[Figure]
* * *
**RC2.25** Line 200 : why not testing the other (lower) frequencies for the wake measurements?

**AC2.25** We had to make a choice between low frequencies and wave frequency. This, and the reason behind our choice, is now explained at the end of Sect. 3:

"*Wake measurements are carried out for motion conditions with wave frequency. These were chosen in place of the low-frequency conditions because the wake*"

*response to wave-frequency motion was not covered by any previous experiments, which instead focused on the low-frequency motion.*"

For sure it would be interesting to carry out the same measurements with low-frequency motion. We included this suggestion in the conclusions:

"*Concerning the wake of floating turbines, suggestion for future work is to carry out measurements with low-frequency motion conditions and measure the wake also in positions further downstream.*"

| | |
|---|---|
| **RC2.26** | Table 3 : a reference to a previous work |
| **AC2.26** | Table 3 gives the power and thrust coefficients of the IEA 15 MW scale model that was developed on purpose for this project and wasn't covered in any previous work. |

| | |
|---|---|
| **RC2.27** | Figure 3 : not clear how the contour lines are calculated |
| **AC2.27** | We think this issue is the same one commented by Referee #1, who noticed that steady-state values in Table 3 do not match the values of Fig. 3. We modified and we included more comments about it, and you can find them in AC1.6. |

| | |
|---|---|
| **RC2.28** | Line 232 : What is IFFT? |
| **AC2.28** | The new results of Fig. 5 do not make use of the IFFT, so we removed the sentence. |

| | |
|---|---|
| **RC2.29** | Line 258 : …the thrust coefficient, … |
| **AC2.29** | Checked. |

| | |
|---|---|
| **RC2.30** | Equation 5 : U_infinity ? instead of U ? |
| **AC2.30** | We prefer to use $U$, and in case of a fixed turbine $U = U_\infty$, whereas for a floating turbine $U$ is equal to the apparent wind speed (i.e., the combination of $U_\infty$ and the velocity created by motion). We reworked the equations 4-7 to make this clearer. |

| | |
|---|---|
| **RC2.31** | Line 285 : "… Van der Veen (2012), where he uses…" |
| **AC2.31** | Checked. |

| | |
|---|---|
| **RC2.32** | Line 286-289 : is this part needed? |
| **AC2.32** | We agree it's not needed, and it is a bit dispersive. We removed it. |

| | |
|---|---|
| **RC2.33** | Line 301 : remove "than" |
| **AC2.33** | Removed. |

| | |
|---|---|
| **RC2.34** | Line 304 : correct the formula |

| | |
|---|---|
| **AC2.34** | Formulas are now correct. |

| | |
|---|---|
| **RC2.35** | Line 311 : "In case of pitch, the phase is not −pi/2 for any frequency" . A bit too straightforward. Elaborate an explanation for that |
| **AC2.35** | Thank you for this comment, that makes the results easier to read and, in the end, more interesting. We added these comments:

"*In case of pitch, unlike what happens with the surge, the phase of the aerodynamic loads measured in the experiment is never −π/2. As it is shown in Fig. 2, blade-level unsteadiness can be the cause of the different phase shift only in the case of pitch motion with wave frequency but is not a valid explanation for cases with low-frequency motion. The rotor reduced-frequency is higher for low-frequency pitch motion compared to surge (2× for the WindCrete and 5× for the Activefloat) and this may give some rotor-level unsteadiness. Another reason for the different phase behavior may be the skewed inflow created by pitch motion. Surge and pitch are equivalent in terms of the apparent wind speed at hub height, but pitch motion produces a velocity gradient across the rotor height, with the upper portion feeling a higher wind speed than the lower one. This is not accounted for by the QST model, which models the rotor as a single point coincident with the hub.*" |

| | |
|---|---|
| **RC2.36** | Line 322 : "…supports the idea that… |
| **AC2.36** | Checked. |

| | |
|---|---|
| **RC2.37** | Line 325 : the blockage effect seems to be very important. |
| **AC2.37** | Thank you again. You are right saying the difference we see is too much to be explained as the effect of blockage. We did some math with the Glauert's model to estimate the overspeed due to blockage and this should be lower than 3%. We think the rest of the difference is due to an offset between the pitot and the hot-wires; however, we don't have measurements without the turbine or measurements with a third instrument to confirm the offset. We included this comment to explain the $U/U_\infty > 1$ at the wake extremities:

"*The velocity at the wake extremities is 17% higher than the free-stream velocity in WC1 and WC2. This difference is partially explained as the effect wind tunnel blockage (according to the model of Glauert (1935) the overspeed due to blockage is 3% for WC1 and 1% for WC2) and most of it may be due to an offset between the hot-wire probes and the upstream pitot tube which is used to take the measurement of $U_\infty$.*" |

| | |
|---|---|
| **RC2.38** | Figure 7 : remove the grey background |
| **AC2.38** | Removed. |

| | |
|---|---|
| **RC2.39** | Conclusion : you do not put anymore the yaw in the class of motion that modify the apparent wind speed |
| **AC2.39** | True, we revised the text in the conclusion in this way: |

> "*Surge and pitch motion move the turbine rotor in the wind direction altering the apparent wind speed.* […] *With yaw motion, the apparent wind is periodically increased on one side of the rotor and decreased on the other, and the apparent wind is higher at the rotor periphery compared to the center.* […] *The main effect of sway and roll motion is to alter the direction of the wind perceived by rotor, whereas the impact on relative wind speed is limited.*"

**RC2.40**   Line 391 : "with focus on…"

**AC2.40**   Checked.

**RC2.41**   Lines 392-394 : this is new information that was not discussed previously. Please add this into the results discussion

**AC2.41**   We included this information in Sect. 4.2.1.

---

## Referee Report (RR1)

L280: Thanks for the effort in improving the paper here. However, it is not clear to me if the rotor speed experienced small variations in operation due to the pitch motion, leading to the inertial effects on rotor torque that we are discussing about, or if the rotor was slipping in the fixed tests without wind. Maybe I would rephrase, attributing the lack of inertia removal to "the inability to completely lock the rotor in the tests without wind". Stating that this depends on the controller is confusing in my opinion, since this way one would assume that a torque controller was used in the tests with wind, while from my understanding the rotor speed is simply held fixed.

Figure 9: I understand this new figure was prompted from the requests of reviewer #2 (and partially from my own). I am not used to analyzing such figures, however despite my best efforts I find this figure extremely difficult to interpret. I will list the points I am struggling with so authors can judge how to possibly improve the figure:

1. I am not 100% sure what the color map refers to. In the sense that: we are looking at average variation with respect to? The mean velocity in each point? Or the mean velocity in a certain point of the phase-Y plane?
2. We are looking at maps derived for oscillations at the wave frequency, correct? Perhaps include this in the legend.
3. Does the rotor-frequency have anything to do with the results we are seeing? Particularly for the results in sway and roll, if no significant meandering can be noted, can the interaction between root and tip vortices of the blades be the cause of the oscillations in the various parts of the wake?

---

## Author Response (AR2)

Politecnico di Milano
Department of Mechanical Engineering
Via La Masa 1, 20156, Milan
Italy

Wind Energy Science Discussion

Date: July 27, 2022
Subject: WES-2022-36 Final Response

Dear Referees,

Thank you for taking the time to revise our manuscript, for the corrections and for the valuable feedbacks.

The article has been revised following your suggestions. The attached document answers your comments.

On behalf of all Authors,
yours sincerely,

Alessandro Fontanella

Attached documents:
- Response to Anonymous Referee #1
- Response to Anonymous Referee #2

**Response to Anonymous Referee #1**

Dear Referee,

Thank you for your additional suggestions. We implemented them in the paper, and you can find below some feedbacks from our side.

- L280: Thanks for the effort in improving the paper here. However, it is not clear to me if the rotor speed experienced small variations in operation due to the pitch motion, leading to the inertial effects on rotor torque that we are discussing about, or if the rotor was slipping in the fixed tests without wind. Maybe I would rephrase, attributing the lack of inertia removal to "the inability to completely lock the rotor in the tests without wind". Stating that this depends on the controller is confusing in my opinion, since this way one would assume that a torque controller was used in the tests with wind, while from my understanding the rotor speed is simply held fixed.

  Thank you for pointing this out. The lack of inertia removal is indeed due to "the inability to completely lock the rotor in the tests without wind". We think this is the best way to express what we meant, so we included the sentence in this way:

  "*The large amplitude of torque variations, which is not matched by thrust, and their phase shift suggest the torque response is mostly due to mechanical inertia which could not be removed by the force post-processing. This is due to the inability to completely lock the rotor in the tests without wind, which resulted in small oscillations of the rotor of less than 10° of amplitude.*"

- Figure 9: I understand this new figure was prompted from the requests of reviewer #2 (and partially from my own). I am not used to analyzing such figures, however despite my best efforts I find this figure extremely difficult to interpret. I will list the points I am struggling with so authors can judge how to possibly improve the figure

  1. I am not 100% sure what the color map refers to. In the sense that: we are looking at average variation with respect to? The mean velocity in each point? Or the mean velocity in a certain point of the phase-Y plane?

     The figure shows the space-time evolution of phase-averaged velocities, and in particular their variation with respect to the mean velocity in each point. We included this information in the caption of Fig. 9. Moreover, we in the text (lines 411-416) a brief explanation of how we obtained the results of Fig. 9.

  2. We are looking at maps derived for oscillations at the wave frequency, correct? Perhaps include this in the legend.

     Yes, it is correct. We included this information in the figure caption.

  3. Does the rotor-frequency have anything to do with the results we are seeing? Particularly for the results in sway and roll, if no significant meandering can be noted, can the interaction between root and tip vortices of the blades be the cause of the oscillations in the various parts of the wake?

Thank you for this comment. We agree this hypothesis is interesting and deserves to be further investigated in future research. We included this sentence in the comment to Fig. 9 at lines 434-436:

"*Since no significant meandering is detected, the velocity oscillations in the central portion of the wake can be the result of the interaction of root vortices that are generated when the turbine operates in WC2*."

And, at lines 444-445 (where results with yaw motion are discussed):

"*As for sway and roll motion, the velocity oscillations in the center of the wake suggest a possible interaction between the root vortices.*"

We also included a sentence in the conclusion (lines 488-490):

"*No significant motion of the wake core is detected, so the velocity oscillation in the center of the wake that are seen with sway, roll, and yaw may be the result of the interaction between root vortices. Further research is needed to confirm this hypothesis and, in general, to explain the velocity fluctuations caused by platform motion.*"

**Response to Anonymous Referee #2**

Dear Referee,

Thank you for your comments and corrections. We fixed the errors at lines 118 and 286, and we updated the caption of Fig. 9 as you suggest.

Finally, concerning the comment:

Line 551 : what is confidential in the Dante calibration procedure? everything is explained in their userguide.

Our answer is that the calibration procedure and the data acquisition system were developed in house and are different from those provided by Dantec (only their LabVIEW drivers are used). This was done by our research group that works on hot-wire probes to improve the performance of the measurement system. The methodology is not published in any paper because it is used also for commercial applications. We briefly explained this in the Appendix B with this text:

"*The calibration procedure and the data acquisition software were developed in house and are confidential.*"